# Sandpile-based model for capturing magnitude distributions and spatiotemporal clustering and separation in regional earthquakes

Rene C. Batac[1], Antonino A. Paguirigan Jr.[1], Anjali B. Tarun[1], and Anthony G. Longjas[2]

[1]National Institute of Physics, University of the Philippines Diliman 1101 Quezon City, Philippines
[2]St. Anthony Falls Laboratory, University of Minnesota, 2 Third Ave. SE, Minneapolis MN 55414, USA

*Correspondence to:* R.C. Batac (rbatac@nip.upd.edu.ph)

**Abstract.** We propose a cellular automata model for earthquake occurrences patterned after the sandpile model of self-organized criticality (SOC). By incorporating a single parameter describing the probability to target the most susceptible site, the model successfully reproduces the statistical signatures of seismicity. The energy distributions closely follow power-law probability density functions (PDFs) with a scaling exponent of around $-1.6$, consistent with the expectations of the Gutenberg-Richter (GR) law, for a wide range of the targeted-triggering probability values. Additionally, for targeted triggering probabilities within the range 0.004-0.007, we observe spatiotemporal distributions that show bimodal behavior, which is not observed previously for the original sandpile. For this critical range of values for the probability, model statistics show remarkable comparison with long-period empirical data from earthquakes from different seismogenic regions. The proposed model has key advantages, foremost of which is the fact that it simultaneously captures the energy, space, and time statistics of earthquakes by just introducing a single parameter, while introducing minimal parameters in the simple rules of the sandpile. We believe that the critical-targeting probability parametrizes the memory that is inherently present in earthquake-generating regions.

## 1 Introduction

The sandpile model, introduced as a representative system for illustrating self-organized criticality (SOC) (Bak, Tang, Wiesenfeld, 1987), has opened up new avenues for the use of discrete cellular automata (CA) models in capturing the salient features of many systems in nature (Olami, Feder, Christensen, 1992; Drossel and Schwabl, 1992; Malamud and Turcotte, 2000; Piegari et al., 2006; Juanico et al., 2008). Seismicity, which is rife with power-law statistical distributions (Saichev and Sornette, 2006), is an interesting test case for such approaches. Despite the complexity of the processes in the earth's crust that limit our ability for accurate, short-term prediction of events, it is worth noting that many statistical features of seismicity, as obtained from substantially complete earthquake records, can be recovered using simple CA models.

One of the earliest attempts for sandpile-based modelling of earthquake distributions is by Bak and Tang (1989), who used a two-dimensional sandpile to show the power-law Gutenberg-Richter (GR) distributions of earthquake energies (Gutenberg and Richter, 1954). Subsequent authors also noted that the simple sandpile produces power-law distributions of earthquake waiting times upon introducing a threshold magnitude (Paczuski et al., 2005). Additional parameters have been introduced

in the model to account for other features of seismicity. Ito and Matzusaki (1990) introduced aftershock triggering to the sandpile model to recover the aftershock frequencies and the hypocenter distributions, which also follow power-law decays. To represent a scale invariant distribution of earthquake faults, Barriere and Turcotte (1991) incorporated a power law distribution of box sizes in the CA model, and recovered not only the GR distribution but the occurrence of foreshocks. On the other hand,

Steacy et al. (1996) investigated the effect of a heterogeneous strength distribution and found that the power-law exponent of the magnitude distribution is dependent on the degree of the heterogeneity. Inspired by the sandpile design, Olami, Feder, Christensen (1992) used a CA implementation of the earlier Burridge-Knopoff model (Burridge and Knopoff, 1967) that incorporates dissipative terms and inhomogeneous energy redistribution rules to capture key elements of seismicity, along with foreshocks and aftershocks (Hergarten and Neugebauer, 2002). In another work, Jagla (2013) has shown that the GR law can

be recovered in a forest-fire model, with the fires interpreted as the earthquake occurrences.

     The introduction of additional parameters to subsequent models indicates that the simplest rules of the original sandpile is not able to capture key features of seismicity. For one, the cascade mechanisms in the grid tend to deplete the stress in the entire system in a single avalanche event, which is in contrast with that of earthquakes where the energy is released in a sequence of correlated events. Additionally, the single triggering at random locations will tend to produce normal distributions of inte-

roccurrence distances and times, which, again, deviates from those observed in records of seismicity. Finally, the conservative sandpile with symmetric nearest-neighbor redistribution rules does not take into account the memory that may be present in actual earthquake-generating zones.

     In this work, we adhere to the key features of the sandpile model, and introduce a very simple modification: for a fraction of the iteration times, determined randomly, we direct the triggering into the most susceptible site in the grid. In this case, the

avalanches in the grid are deemed to be analogous to the energy release during an earthquake occurrence. Interestingly, this very simple modification in the sandpile rule enabled us to recover, simultaneously, the distributions of event sizes, interevent distances, and interevent times that are comparable to those obtained from substantially complete earthquake records.

## 2   Model Specifications

The model utilizes a two-dimensional space discretized into a grid of $L \times L$ cells arranged in a square lattice. The cells contain

continuous-valued information states $\sigma$ representing the local measure of susceptibility to rupture. At time $t = 0$, the states are initialized to have values within $[0, \sigma_{\max})$, where, in this case, we set $\sigma_{\max} = 1.0$ as the relative measure of the rupture threshold.

     The dynamical evolution of the grid is guided by rules patterned after the Zhang sandpile that uses continuous-valued states (Zhang, 1989). We choose an asynchronous update rule, such that every discrete time step, the grid is triggered by

adding a constant value $\nu$ to a single location $(x, y)$, $\sigma(x, y, t + 1) \rightarrow \sigma(x, y, t) + \nu$, with all the other sites unperturbed. The asynchronicity may represent the nonuniformity of crustal motion that drives the accumulation of elastic potential energy at faults. Moreover, the model introduces a targeted triggering probability $p$ that the most susceptible site, i.e. the site with the highest $\sigma$ value in the grid, will receive the driving term $\nu$. Triggering is therefore applied to the most susceptible site with

probability $p$ and to a randomly chosen site with probability $1-p$. The value of $p$ represents a memory term, and parameterizes the tendency of fracture to occur at more susceptible locations along an earthquake generating zone.

In the event when a cell matches or exceeds a maximum possible value $\sigma_{\max}$, the local region is deemed to rupture. No new trigger is added to the system during such events; instead, the stress from the collapsing site is transferred to the four nearest neighbors in the grid, $\sigma(x\pm1,y\pm1,t) \to \sigma(x\pm1,y\pm1,t) + \frac{1}{4}\sigma(x,y,t)$, leading to the relaxation of the original site, $\sigma(x,y,t) \to 0$. Such relaxations may produce a cascade of subsequent stress redistributions and relaxations in the grid when one or more of the neighbors are themselves driven to the threshold. As in the previous sandpile models, the number of affected sites in the grid, $A$, is tracked to quantify the relative event size. Additionally, we also recorded the number of unique activations $V$, the number of times a cell has been affected by a cascade, as a proxy for the actual energy or seismic moment of the relaxation event.

Prior calibrations show that $\nu = 10^{-3}$ produce power-law event-size distributions comparable to the GR law, and that $t_{\max} = \{1, 4, 16\} \times 10^7$ iterations, where the first $10\%$ are neglected for transient behavior, produces substantial number of avalanche events for $L = \{256, 512, 1024\}$ grids, respectively. We investigated the case of different targeted triggering probabilities $p = \{0, 1\times10^k, 5\times10^k, 1\}$, where the integer $k$ is from -5 to -1 to scan a wide range of possible system behaviors. For each of the $p$ values, we track all nonzero $A_i$ and $V_i$ and their avalanche origins and occurrence times $(x_i, y_i, t_i)$, where $i$ denotes the temporal index of occurrence of an event. The spatial and temporal separations of successive events, $R_i = [(x_i-x_{i-1})^2+(y_i-y_{i-1})^2]^{1/2}$ and $T_i = t_i - t_{i-1}$, are computed, and the probability density functions (PDFs) of all $A$, $V$, $R$, and $T$ are plotted.

Records of very low-magnitude earthquakes are oftentimes incomplete, because they are both too weak for detection and because their occurrence is orders of magnitude in frequency as compared with the higher-magnitude ones. In the model, however, we can resolve all the avalanches, even the smallest ones that affect only single neighborhoods. To mimic the effect of the non-retention of the smallest earthquakes, we employed a thresholding procedure in the analyses by setting $A_{th} = \{5, 10, 50, 100, 500, 1000, 5000\}$ such that all events with $A < A_{th}$ are removed from the sequence. Because the $A$ PDF is just expected to be cut off below $A_{th}$, we observed how the statistical distributions of $R$ and $T$ will be affected upon employing different $A_{th}$ values.

Finally, as a way of comparison and verification, we compare the model statistics with those obtained from actual earthquake catalogs from Japan (JP), Philippines (PH), and Southern California (SC), as investigated in a previous work by Batac and Kantz (2014). The JP records are obtained from the Japan University Network Earthquake Catalog (JUNEC), with approximately 137,000 events from July 1985-December 1998; the PH earthquakes are composed of 70,000+ events from 1973 to 2012, as obtained from the Preliminary Determination of Earthquakes (PDE) Catalog; while the SC records are from the Southern California Earthquake Catalog (SCEC) containing 516,000+ events from 1982 to 2012 (events due to man-made activities are removed). We compared the behaviors of the model and data statistics using scaling factors derived from model parameters.

## 3 Model Results

Figure 1(a) shows the avalanche size PDFs for the different values of the targeted triggering probability $p$. For the broad range of $p$ values considered, the distributions are found to be comparable to a power-law $A^{-\alpha}$ with $\alpha = 1.6$. Continuous-state sandpiles have been known to have avalanche size scaling exponents greater than 1.0, the exponent of the discrete Bak-Tang-Wiesenfeld (BTW) sandpile. Lübeck (1997) conducted large-scale simulations of a similar Zhang sandpile and obtained exponents slightly higher than 1.2, which can go even higher for large driving rates $\nu$. In a similar model that incorporated non-conservation, Piegari et al. (2006) obtained power-law exponents that approach 1.6 in the conservative limit for the same order or magnitude of $\nu$ that we used. The higher exponents and the effect of the driving rates are also verified by an equivalent conservative model and actual sand avalanche experiments by Juanico et al. (2008), and in other asynchronous updating models (Paguirigan et al., 2015).

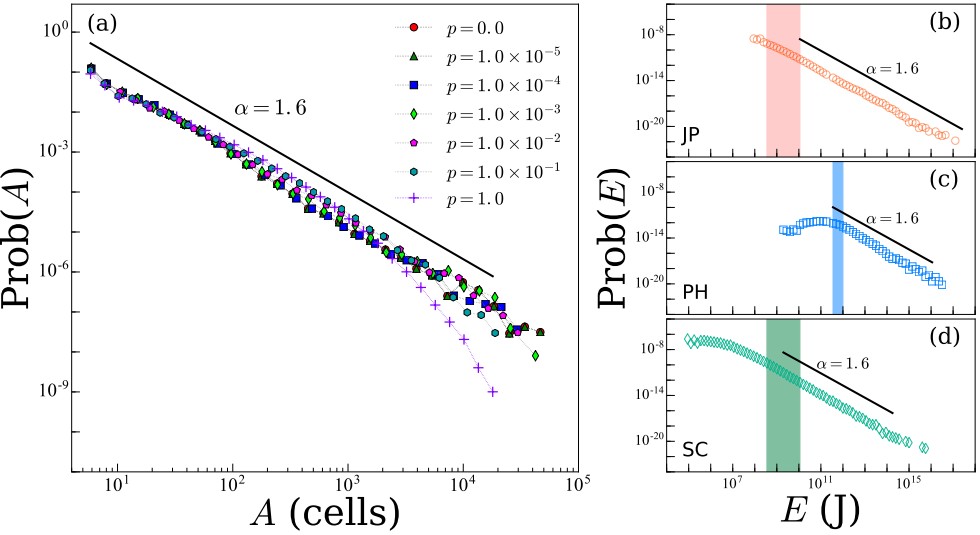

**Figure 1.** Avalanche size and earthquake energy PDFs. For all figures, lines corresponding to the power-law trend with exponent $\alpha = 1.6$ are provided as guides to the eye. (a) Model results show similar behaviors despite the large differences in $p$, signifying the retention of sandpile characteristics. The obtained power-law distributions are comparable to the power-law trends in the energy distributions from (b) Japan, JP; (c) Philippines, PH; and (d) Southern California, SC. In (b)-(d), the horizontal axes scales are preserved; shaded regions denote energy values with substantial completeness, which will be used for subsequent analyses.

The resulting power-law exponent is deemed to be a result of the accumulation of stress at various locations: because the triggering is done at only a single site every time, there is little global connectivity among critical sites, resulting in a preponderance of smaller, isolated avalanches. The fact that the distributions are almost similar regardless of the value of $p$ indicates that the targeted triggering probability has minimal effect on the avalanching mechanism of the grid, such that the system preserves the SOC characteristics of the original sandpile. In contrast, the OFC model, for example, tends to lose the universality of the exponents upon the introduction of nonconservation (Olami, Feder, Christensen, 1992).

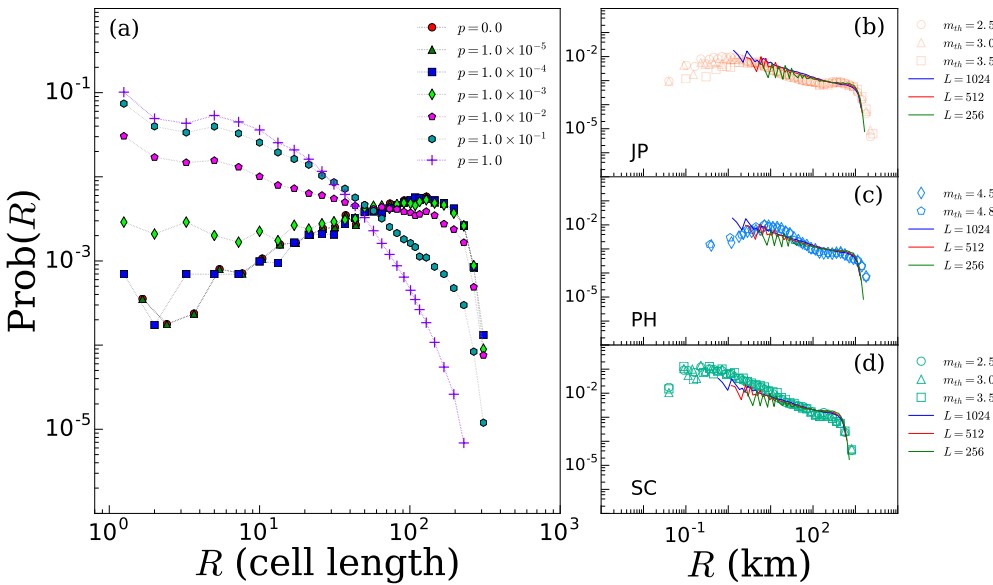

**Figure 2.** (Color online) Interevent distance statistics of model, with rescaling for comparison with actual earthquake separation distance data. (a) For a $L = 256$ grid, higher $p$ results in the preponderance of short-$R$ values. The trends of the model closely mimic those of the data for (b) JP, (c) PH, and (d) SC, where calibration was done by comparing the modes of the model $p = 0$ and shuffled sequences of the empirical data. Larger grids in (b)-(d) result in the capability to replicate the shorter $R$ regimes.

    In Figure 2(a), we observe that the original sandpile $p = 0$ produces unimodal statistics, whose tails decay towards the largest possible distance $\sqrt{2}L$ in the finite grid. The simple sandpile, therefore, is not capable of replicating the observed earthquake separation distance distributions, which are found to exhibit bimodality due to the difference in the characteristic times of the correlated aftershock sequences and the independent mainshocks (Baiesi and Paczuski, 2004; Zaliapin et al., 2008; Touati et
al., 2009; Batac and Kantz, 2014). This inspired the introduction of $p$, which is a random occurrence in time but is inherently affecting the spatial distribution of events in the grid. We do note here that the parameter $p$ is just the probability to target the most susceptible site in the lattice, unlike previous implementations that actually pre-select the next targeting location within the vicinity of the previous avalanche (Ito and Matzusaki, 1990). Indeed, without the imposition of such a spatial bias, the replication of the short-$R$ regimes is not guaranteed. Interestingly, however, the plots in Figure 2(a) show increased probability
of occurrence of the short-$R$ distances upon introducing nonzero $p$. From this, we can deduce that the most susceptible sites in the lattice are most likely to be found within the vicinity of a previous large avalanche, a fact that was not exploited by earlier similar models. In fact, in the biased case $p = 1$, we recovered unimodal statistics, as shown in Figure 2(a), albeit at a shorter characteristic distance; for the $L = 256$ grid, the average location of the most susceptible site from the previous avalanche origin was obtained to be around 21 cell lengths. Midway between these two extremes ($p = 0$ for the original, and $p = 1$ for
the completely biased sandpile), we can find a suitable value of $p$ where reasonable comparison with empirical data can be obtained.

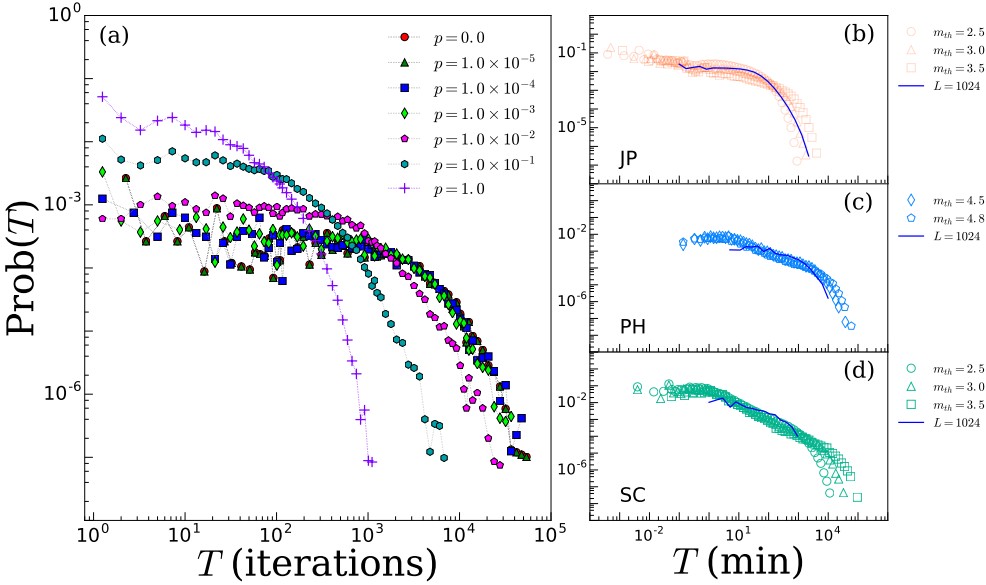

**Figure 3.** (Color online) Interevent time statistics of model, and rescaling for comparison with actual earthquake waiting time data. (a) For a $L = 256$ grid, higher $p$ results in the shift of the distribution to shorter $T$ values. To obtain substantial power-law regimes, we used the results for the $L = 1024$ grid to replicate the waiting time statistics of (b) JP, (c) PH, and (d) SC. By preserving the fraction of events left upon imposing thresholding, we obtained $A_{th}$ values of (b) $5 \times 10^3$ for JP; (c) $5 \times 10^5$ for PH; and (d) $5 \times 10^3$ for SC. The shortest waiting times in the data are scaled to be a unit of iteration. The finite total iteration times resulted in model distributions that are not able to capture the very long tails of those of the empirical data, especially for (d) SC, which has the longest period among the catalogs considered.

The interevent time distributions are shown in Figure 3(a), for $L = 256$ and $t_{\max} = 10^7$ iterations. We observe the expected shift of the tail cutoff towards shorter $T$ values as $p$ is increased; triggering the highly-susceptible sites will more likely result in a new avalanche event, thereby shortening the average waiting time. The resulting distributions are for the case wherein all the events are included in the sequence; we expect a lengthening of the tails of the distributions when we neglect other events below the threshold $A_{th}$.

## 4   Discussion

### 4.1   Energy Distributions and the Gutenberg-Richter Law

The GR law, which is usually presented in terms of the magnitudes $m$ and as a complementary cumulative distribution function (CCDF), $\log_{10} m = a - bm$ can be shown to be equivalent to an energy $E$ CCDF that behaves as $E^{-2/3}$ from the definition of $m$ and by assuming $b = 1$, which is the case for most complete records (Jagla, 2013). By noting that the CCDF is effectively an integral of the PDF, the earthquake energy PDF will then behave as $E^{-5/3}$. In Figure 1(b)-(d), similar power-law trends have been obtained for the JP, PH, and SC records, which have different levels of catalog completeness, as indicated by the

extent of the power-law regimes. To minimize the problems associated with the inherent incompleteness of smaller-energy events (Zaliapin and Ben-Zion, 2015), we impose a threshold magnitude $m_{th}$ for succeeding analyses such that earthquake events with magnitudes lower than $m_{th}$ are dropped from consideration. The range of such magnitudes considered, which are well within the power-law regimes of the plots, are shaded in Figure 1(b)-(d): $m_{th} \in [2.5, 3.5]$ for JP and SC and $m_{th} \in$ 5   $[4.5, 4.8]$ for PH.

In keeping with the earlier sandpile-based approaches where the avalanche size $A$ is used for comparison with earthquake energies (Bak and Tang, 1989; Ito and Matzusaki, 1990), we present in Figure 1 the PDFs of $A$ with those of $E$ from the seismogenic regions considered. It is worth emphasizing that similar power-law trends result from the introduction of the parameter $p$, regardless of how large its relative value is. We note, however, that aside from the avalanche size $A$, there are 10   other parameters that can be used to track the extent of the avalanche event. One such measure is the number of activations $V$, wherein the sites repeatedly affected by the avalanching process gets to be counted multiple times. Previous works have shown that $V$ and $A$ in discrete models may in fact have actual associations with the seismic moment and fracture area, respectively, and may exhibit nontrivial scaling relations (Landes and Lippiello, 2016). We present in Figure 4(a) the distributions obtained upon tracking $V$.

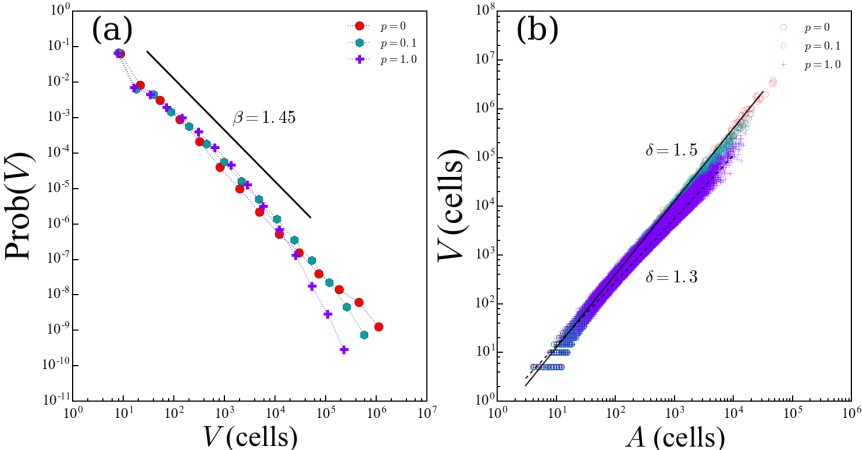

**Figure 4.** (Color online) Model statistics for $V$, and scaling with $A$. (a) The PDF of $V$ (shown here with a power-law $V^{-\beta}$ where $\beta = 1.45$ as guide to the eye) also shows minimal changes upon introducing $p$. (b) The scaling behavior of $V$ with $A$ is supralinear, with behaviors ranging from $V \propto A^{1.5}$ for $p = 0$ to $V \propto A^{1.3}$ in the regime of large $A$ values for $p = 1$.

15   The $V$ PDFs also show a behavior similar to those of their corresponding $A$: there are minimal changes upon scanning for different $p$ values. The distributions also follow power-law behaviors $V^{-\beta}$ with $\beta$ around 1.4 to 1.5 (the case of $\beta = 1.45$ is plotted as a guide to the eye in Figure 4(a)). The parameter $V$ is a better representation of the energy $E$ in earthquakes, and the obtained scaling exponent $\beta$ is still deemed to be close to the earthquake energy scaling exponents. The fact that the model can replicate the energy statistics is a vital first requirement for any discrete model of earthquakes. Additionally, the preservation

of the power-law exponent for almost any value of $p$ indicates that the model does not deviate significantly from the original sandpile behavior, and may exhibit (self-organized) criticality.

To understand the scaling relations between $V$ and $A$, we plot the $V$ (activated cells) vs. $A$ (affected cells) in Figure 4(b) and note that the scaling relations, which are higher than linear, change for higher $p$. The case of $p = 0$ (randomly-triggered sandpile) results in a $V(A) \propto A^{1.5}$ scaling. On the other hand, for $p = 1$ (sandpile with targeted triggering), the behavior appears to shift towards $V(A) \propto A^{1.3}$ for very large $A$ values. This lower scaling exponent of the activation for large avalanche sizes is expected for targeted triggering; because the most susceptible site is always targeted, there is minimal accumulation of near-critical sites near the location of the avalanche origin, which results in lower number of reactivations of affected sites near an avalanche event.

## 4.2 Spatial Separation of Earthquake Events

In the original asynchronous sandpile models, one only recovers unimodal statistics for interevent distances. This is due to the stochastic nature of the triggering: the next location to be perturbed is drawn from an oftentimes uniform distribution, i.e. all sites are likely to be triggered next. Additionally, the nature of internal cascading within the sandpile grid results in the depletion of all the critical sites within the extent of the avalanche area. The same cannot be said of earthquakes: after the release of elastic potential energy at a fault location, the subsequent crustal motion may tend to favor other fractures near the vicinity of the earlier event, to release the remaining stored energy.

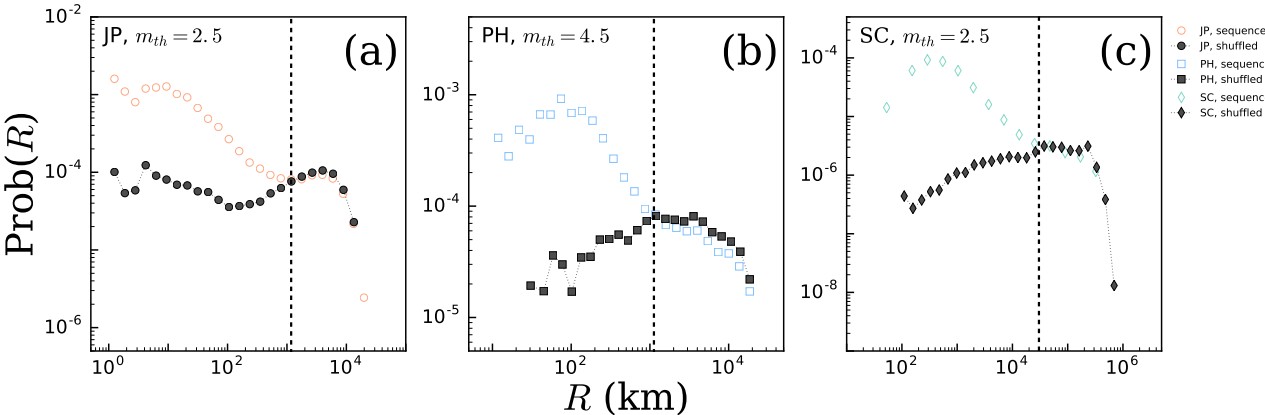

**Figure 5.** (Color online) Empirical earthquake interevent distance distributions [hollow symbols], along with the corresponding shuffled sequences [filled symbols] for (a) JP, (b) PH, and (SC). The broken lines indicate the $R^*$ values where the original and the shuffled sequences begin to show similar trends.

Interestingly, the addition of the simple targeted triggering probability $p$ have enabled us to recover statistical distributions that are comparable to those observed in regional earthquake records, up to a scaling factor. It should be noted that without any form of spatial clustering, the characteristic separation distance is limited by the finite system size. Rescaling is therefore

conducted by comparing the characteristic sizes (modes) of the memoryless cases of the model ($p = 0$) and the data (shuffled sequence). The interevent distance distributions of the shuffled sequences are shown as the black symbols in Figure 5, while the corresponding model $p = 0$ distribution is shown in Figure 2(a), with both clearly showing unimodal statistics.

Upon getting the rescaling factor, we scan through the possible $p$ values to obtain $p$ values that will result in comparable $R$ distributions between model and data. We observe that the model parameters that will correspond to the empirical distributions upon such a simple rescaling ranges from $p^* \approx 0.004 - 0.007$. Figure 2(b)-(d) shows the interevent distances between successive earthquakes in the different regional records considered, superimposed with the rescaled statistics of the model.

The rescaled model statistics for $p = 0.007$ show good agreement with interevent distances from the three seismogenic regions. As expected, larger grid sizes will result in a better discrimination of shorter $R$, i.e. one pixel unit will correspond to shorter actual distance units. In our case, for the largest grid size used ($L = 1024$), we find that the scaling factors obtained by matching the modes result in the following correspondence with a unit cell length: 1.3 km for JP, 1.2 km for PH, and 0.5 km for SC. The distributions are found to be similar regardless of the threshold magnitude $A_{th}$ considered due to the finite system size; even upon removing the weakest events, the avalanche origins are confined within the grid, resulting in the same $Prob(R)$.

## 4.3 Temporal Separation of Earthquake Events

The temporal separation of aftershocks and mainshocks that have different characteristic waiting times is an intuitive result that is both well-known and widely studied (Zaliapin et al., 2008; Touati et al., 2009; Batac and Kantz, 2014; Batac, 2016). The proposed model, therefore, must also show these features to be able to compare reasonably well with the temporal distributions of seismicity. In the following, we compare the results of the model having $p^* = 0.007$ and grid dimension $L = 1024$, which has been shown to have comparable $R$ statistics with empirical data.

In comparing model and empirical temporal interevent statistics, one does not have the similar advantage of having a finite "space." The goal of rescaling in time is to recover the relatively short $T$ regimes first; theoretically, the longest $T$ will be recovered if the model is allowed to run for very long iteration times. Additionally, in rescaling the time, one should take into account the fact that the earthquake record is thresholded by $m_{th}$, effectively lengthening the average time between the occurrence of two events. Ideally, if all the events, no matter how weak, can be detected and recorded, we would not have long tails in the waiting time distribution of earthquakes. This is also observed in sandpile-based models; previous approaches have shown that the waiting time distribution will be Poisson distributed when all the events are considered, but will begin to show apparent power-law characteristics upon thresholding (Paczuski et al., 2005; Juanico et al., 2008).

For our purpose, we arbitrarily chose the following threshold avalanche sizes for removing weaker events: for comparison with JP and SC, which are both taken to have $m_{th} = 2.5$, we used $A_{th} = 5 \times 10^3$; on the other hand, for PH, with relative completeness beyond $m_{th} = 4.5$, $A_{th} = 5 \times 10^5$ is used. The values of $A_{th}$ are obtained by maintaning the fraction of events left after neglecting the weaker events. Still, because of the limited number of regional data sets considered that does not allow for further testing their correspondence, we emphasize that the values of the $A_{th}$ obtained does not necessarily translate into an exact equivalence with the threshold magnitude $m_{th}$ for the data.

Upon removing the events with $A < A_{th}$, we obtained the modes of both the data and the model for visual comparison. This resulted in slight differences in the rescaling factors for the different data sets. One iteration of the model corresponds to: 0.006 s for JP; 0.004 s for PH; and 0.002 s for SC. Figure 3(b)-(d) above show the rescaled model distributions alongside the those of the empirical data, showing qualitative similarities in their trends.

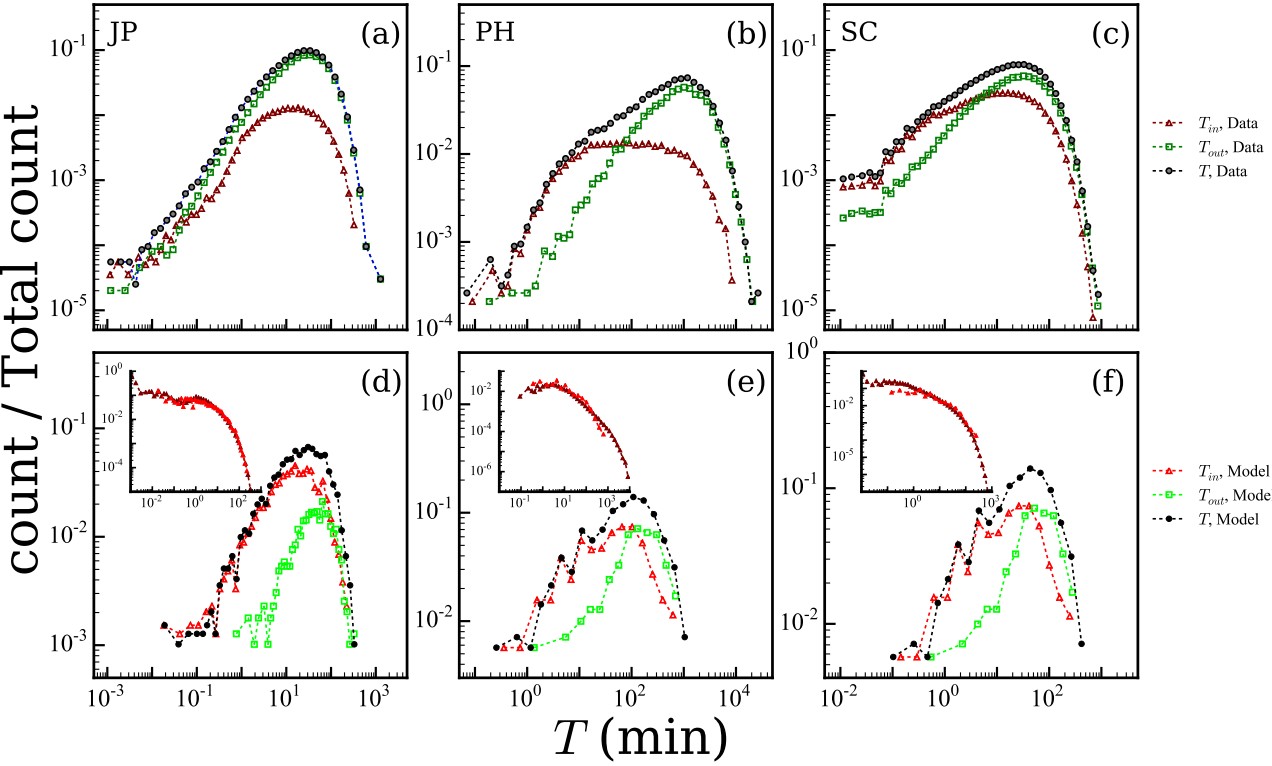

**Figure 6.** (Color online) Conditional relative frequency distributions of $T_{in}$ and $T_{out}$ for (a)-(c) earthquake data and (d)-(f) corresponding rescaled model results, plotted with the relative frequency plot of all $T$. Nearby (Far away) events have higher (lower) chance of having short waiting times and lower (higher) chance of having long waiting times, as can be seen from the modes of the conditional frequency distributions. The insets of (d)-(f) show that the $T_{in}$ PDFs of model and rescaled data have significant overlap, signifying the similarities in their correlated origins.

Apart from recovering the qualitative trends in the $T$ PDFs, we conducted additional analyses to check if the model results also show spatiotemporal clustering and separation behaviors. In Figure 5, we mark the location of $R^*$, the characteristic separation distance where the empirical distributions and those of the shuffled sequences begin to show comparable trends. The $R^*$ values of the region considered, which is similar to the results of Batac and Kantz (2014), is deemed to be a good marker for separating "nearby" and "far away" events. We note that a similar procedure done using the rescaled model statistics results in comparable $R^*$ values. Using $R^* = 164$ km for JP, $R^* = 125$ km for PH, and $R^* = 79$ km for SC, we separated the

corresponding waiting times $T$ into the sets $T_{in} = \{T|R \leq R^*\}$ and $T_{out} = \{T|R > R^*\}$. Figure 6 shows the relative frequency plots of $T$, superimposed with those of $T_{in}$ and $T_{out}$, for the empirical data and the rescaled model values.

As shown in Figure 6(a)-(c), for all the seismogenic regions considered, the distributions of $T_{in}$ and $T_{out}$ differ significantly from that of the total $T$. The relative frequency plots of $T$ in all cases can be shown to be a crossover between $T_{in}$ and $T_{out}$

that have different modes. As expected, the $T_{out}$ distributions do not coincide due to the different periods involved in the catalogs considered. The $T_{in}$ distribution, on the other hand, all show modes at short $T$ values, suggesting a strong dependence among the interevent properties in space and time (Livina et al., 2005). This conditional distribution therefore quantifies the spatiotemporal clustering observed in earthquakes, particularly among aftershock sequences that result from the correlated mechanisms: "nearby" events are also more likely to be separated by shorter waiting times.

In Figure 6(d)-(f), we observe that despite the shorter iteration times being considered, the model was able to show the separation of the $T_{in}$ and $T_{out}$ distributions, a feature that is also found in empirical data (Batac and Kantz, 2014) and in other earthquake models (Touati et al., 2009). Moreover, it is particularly interesting to note that the rescaled $T_{in}$ statistics of model and corresponding $T_{in}$ from the earthquake data show comparable trends, especially for shorter waiting times, as shown in the insets. The $T_{in}$ statistics has been shown to correspond with the statistics of aftershocks, as shown in studies of fresh aftershock

statistics from empirical data (Batac, 2016). This suggests that the correlated mechanisms in actual earthquake systems that produce the $T_{in}$ distributions are also present in the model.

### 4.4 Model Advantages and Insights on Empirical Modeling

Introducing the parameter $p$ into the sandpile driving is a straightforward way of incorporating memory into the system. This simple parameter holds a distinct advantage over other models that introduced additional parameters, because it spans a

wide range of possible statistical distributions in event size, space, and time, without actually biasing the location of the next triggering event. Being a single parameter, the correspondence between $p$ and actual properties of the earthquake-generating system may be difficult, if not impossible, to ascertain. At best, we may think of $p$ as a combined effect of many different factors on the ground that lead to the preferential triggering of a location.

We believe that this parameter, which, for earthquakes, show comparable statistics for the range $p^* \approx 0.004 - 0.007$, may

be introduced in other sandpile-based models of other events in nature deemed to be showing self-organized (critical) characteristics. It may be possible to quantify the extent of "memory" of these systems through the value of the parameter $p$ that best replicates their statistical distributions.

Moreover, a deeper analysis of the other regimes of $p$ may lead to a better comparison between the model and other similar protocols. For example, for higher values of $p$, the model may exhibit extremal dynamics, resulting in more avalanche events

due to the tendency to always trigger the most susceptible site. On the other hand, for very low values of $p$, the dynamics may be comparable to other models that employ uniform loading. Knowing these limits, and establishing how similar and/or different the model is from other discrete models may help put the results in a better context.

# 5 Conclusions

In summary, we have presented a simple cellular automata model inspired by the original sandpile model. The model avoids introducing biased rules, and instead incorporates a probability of targeting the most susceptible site in the grid, reminiscent of the assumed fracture mechanism of actual earthquake systems. Within a small range of values $p^* \approx 0.004 - 0.007$, we have observed that the model statistics show comparable trends with empirical distributions of earthquake occurrences in energy, space, and time, upon simple rescaling.

The work has also uncovered an important property of the sandpile grid: the most susceptible sites lie within the vicinity of a previous large avalanche event. Previous sandpile-based models that synchronously update all lattice sites, or those that asynchronously update at random locations, are not able to exploit this important property, preventing the possibility of directly modelling earthquakes using the sandpile paradigm. The introduction of such a targeting probability without destroying the sandpile properties may hint at self-organized critical mechanisms at work in the grid. The fact that the simple targeted triggering probability simultaneously recovers these important statistical features of earthquakes is a simple yet novel concept that has not been exploited by previously-proposed discrete models based on the sandpile.

Deeper analyses and comparisons with other established models of seismicity may help further establish similarities and differences and put the model results in a better context. Additionally, the parametrization of memory in the form of the targeted triggering probability may be extended to other similar models to possibly capture the statistical distributions of other self-organized (critical) events in nature and society.

*Author contributions.* R.C.B. devised the model and A.A.P.J. run large-scale simulations. R.C.B. and A.B.T. wrote the paper. A.G.L. and A.B.T. provided the empirical data and model comparisons. A.B.T. and A.A.P.J. conducted statistical goodness-of-fit tests.

*Acknowledgements.* The authors would like to acknowledge financial support from the University of the Philippines Diliman (UPD) Office of the Vice Chancellor for Research and Development (OVCRD) through a PhD Incentive Award with project title "Quantifying the clustering characteristics of complex, self-organizing systems in nature and society." A.A.P.J. acknowledges the Department of Science and Technology (DOST) for his Advanced Science and Technology Human Resources Development Program (ASTHRDP) scholarship.

We extend our gratitude to R. Gloaguen (editor), F. Landes, S. Hergarten, and one anonymous referee for recommendations that significantly improved the content and the presentation of the manuscript.

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
