# Peer review of "Sandpile-based model for capturing magnitude distributions and spatiotemporal clustering and separation in regional earthquakes"

_Nonlinear Processes in Geophysics, 2016_

## Referee Comment (RC1) · François P. Landes (Referee) · 4 Jul 2016

It stroke me at the first read, how much the context and references to previous or related works were lacking. There are few references to recent works related to sandpiles and other similar (lattice, mesoscopic) models. I think the authors should not limit their readings to models that explicit mention sandpiles or OFC in their titles, as there is a whole family of other models which share many interesting properties, and are morally very similar, if not exactly mapped to some particular variants of sandpiles.

In this respect, I am not completely sure that the results presented here are completely novel. They are probably novel to some extent, but should be compared with existing, close-by literature.

Overall, the paper is not badly written, but there are important things that are not easy to grasp, like the precise definition of how things were fitted or "calibrated", and the definitions of the quantities plotted in the figures. I detail what I mean by this in the points below.

0) About context:

The following forest-fire model should, I think, be put in perspective with your work, as it shares some of its ingredients: "Forest-Fire Analogy to Explain the b Value of the Gutenberg-Richter Law for Earthquakes".

More recently, the same author (and others, see references inside or citing it), did a paper dealing precisely with how the loading protocol affects the statistics of events in plasticity/fracture: "Avalanche-size distributions in mean-field plastic yielding models". That one is less tightly connected to your work, but it compares two kinds of driving (random and uniform), and it belongs to the family of models using "extremal dynamics" (loading all sites until exactly one is triggered).

In a sense, your protocol is quite similar to doing:

- uniform loading at most steps.

- with small proba p, trigger the most susceptible site by adding to it the needed stress (and maybe add the a fraction of that amount of stress to every other site). (in extremal dynamics, one adds the same amount of stress to all sites)

I think it would actually increase the impact of your paper to connect your protocol and results to these other existing protocols and results.

You may want to check out, also, and see for yourself how relevant the following papers are:

-"Universality in Sandpiles, Interface Depinning, and Earthquake Models", Paczuski et al (1996).

-"Avalanche size distributions in mean field plastic yielding models"

- (about SOC, 1 example:) "Dynamic scaling in stick-slip friction" (2005)

- "Strain localization and anisotropic correlations in a mesoscopic model of amorphous plasticity", where "extremal dynamics" is used (there are countless others, not every author focuses on the dynamics and uses these words).

Aside from the context-building, other remarks:

1) Several times, you mention the similarity with the original sandpile model as a good thing, in particular the fact that the model is SOC seems to be very positive. For instance in the introduction, you say: "make the model more truthful to the original sandpile design, presenting a clear association with seismicity and SOC.". Why do you consider SOC and sandpiles to be a good thing by itself?

As Fisher noted in his 1998 Review: "Whether critical behavior is considered "self-organized" or not is somewhat a matter of taste: if the systems we are considering are driven at very slow velocity, then they will be very close to critical. In another well known situation, when a fluid is stirred on large scales, turbulence exists on a wide range of length scales extending down to the scale at which viscous dissipation occurs. In both of these and in many other contexts the parameter which is "tuned" to get a wide range of scales is the ratio of some basic "microscopic" scale to the scale at which the system is driven."

I don't think I could explain better than him: SOC is nowadays, to many people, not a particularly relevant characteristic. Often, what is called SOC is just a model where the critical point is at 0 or infinity. In your case, it is the system size which acts as a limiting size for the avalanche (dissipation occurs only at the boundaries in your model, not in the bulk, if I understood well.)

By the way, your driving depends on the system's state, which in a sense can be seen as a feedback loop... thus giving a weaker "self-"organized structure to the problem.

I think this perspective on SOC and sanpiles should be updated/deleted, as they do not seem to bring anything to the paper. If you persist in liking SOC so much, you should give some explanation of why it is such a good thing that your model has some SOC in it.

2) p.3, line 8:

You say: "The number of affected sites in the grid, A, is used as a proxy for the actual energy ...". I think you should count the number of activations, not the number of sites activated once or more. If some sites are activated twice (or more), they should count twice (or more). Given that you put no dissipation, I guess it can occur ... maybe often?

If this is indeed what you measured, be more clear.

If you did not, you should show both quantities. The number of sites activated (irrespective of the number of activations) represents the area of EQs. The number of activations represents the seismic moment (energy released).

3) same place:

is used as a proxy for the actual energy or magnitude of

replace "magnitude" with "seismic moment", as only this quantity has the dimension of an energy. Magnitude is the log (up to a prefactor) of the seismic moment.

4) A general, important criticism: it is not clear how much you calibrated to get results to fit experimental data. More precise, yet clear explanation/discussion of the number of degrees of freedom (fitting parameters) in your fits, would be welcomed. Otherwise, the whole point of the paper (i.e. that your fits are rather good and relying on few fitting parameters) is compromised.

In this respect, I find the figures not very clear. In general, the methods should be clearly explained. Using a few more explicit sub-titles along your presentation may help.

5) Related to my point (1): discuss also maybe, how often (what proba) is it that the site triggered was the most susceptible site of the state after the previous avalanche? By this I mean, if you record the position of the most loaded site after an event, how often is it that the triggering site of the next event is precisely the same site?

I suspect this is much larger than p.

I think it is good to discuss this, as it is a natural question the reader may have, and it could help relate your model to others.

6) By the way, in your model, is the area VS energy scaling in some way? These features are expected to have multiple scaling behavior, is it the case in your model?

I recently published a study of various models, discussing this particular observable as a benchmark of model's quality, i.e. the area-magnitude scaling relationship: "Scaling laws in earthquake occurrence: Disorder, viscosity, and finite size effects in Olami-Feder-Christensen models"

This can be a tricky thing to study, and the fact you do not recover expected natural-data behaviour for this observable does not discard your model as uninteresting. I am just suggesting this as possible directions for future work.

7) figures: do not write PDF, but rather P(A), P(E), P(T), etc. (or Prob(A), etc, as you prefer). It would be more clear.

8) "... observed in the generation of earthquakes, which, despite regional differences, produce universal GR distributions.". This statement is rather controversial, and should be supported at least by a citation. One should not be overly confident with Per Bak's statements, (which were overly enthusiastic about SOC... and sometimes plainly wrong).

Geophysicists are quite interested in knowing the GR law region by region. It has regional differences, and integrating over all regions does not necessarily carry lots of physical meaning.

9) "However, for the threshold Ath=50 used, we have not seen the power-law regimes due to the ..."

I did not understood the definition of quantity A_th. Is it given somewhere? If not, give it. If so, make it more visible.

10) "In Figure 2(b)-(d), the we find that the rescaled model statistics for p=0.007 show good agreement". Correct the typo.

---

## Referee Comment (RC2) · Anonymous Referee #2 · 5 Jul 2016

I think that overall the article is well written and the presented results are sufficiently clear, even though some passages are poor of references and some of the methods that were used are not fully explained.

In more detail:

1] The authors refer to the Zhang sandpile, introducing a deterministic toppling rule (the stress from the toppling site is equally redistributed between its nearest neighbors). It is known that the deterministic sandpile exhibits anomalous multi-scaling because of the breakdown of ergodicity caused by the existence of many toppling invariants [Bagnoli et al. 2003 Europhys. Lett. 63 512]. (If it isn't too annoying) I would suggest to choose a random toppling rule instead (that guarantees the model to belong to Manna

universality class).

2] Why is the exponent of the avalanche size probability density functions ($\alpha=1.6$) so different from the exponent found for 2-dimensional sandpiles under synchronous updating rule ($\alpha=1.26$) -even in the limit of p=0- ? I think that this should be discussed or some literature should be cited. The only cited article with respect to this issue is Paguirigan et al., 2015, which deals with the introduction of sinks leading to non-conservation. I would suggest to clarify how does this relate to asynchronous updating rule. The other cited article (Lubeck, 1997) compares the static and dynamical properties of Zhang sandpile (the same that is considered here) with those of the Abelian sandpile model of Bak, Tang, and Wiesenfeld, stating that the exponents of the avalanche probability distribution are the same. I think that some more (and more relevant) references should be cited here, in order to give stronger evidence for the appearence of the exponent 1.6 (given that it appears to fit really well with the experimental data).

4] In Figure 2 (b)-(d) I guess that black dots represent shuffled data but I think that it should be written explicitly for better readability.

5] The authors state and show in Fig. 2 that the distribution of the spatial distances between events becomes bimodal for some values of p, stating that "bimodality [in the experimentally observed distributions] is due to the difference in the characteristic times of the correlated aftershock sequences and the independent mainshocks". I think it would be useful that the authors discuss why such distribution becomes bimodal in the model, for some intermediate values of p.

6] I think that the "calibration with real-world data" procedure is not perfectly clear. I would suggest to explain the procedure in more detail. As it is, I can imagine it is only a way to plot simulations results and experimental data on the same plot, in other world it is an artifact that allows to compare quatitatively two data sets that in principle can only be compared qualitatively but that does not explicitly add any new information to

the article.

7] In Line 3 page 6 the authors write: "The GR law [...] can be shown to be equivalent to an energy E CCDF ...". I think it would be useful to cite here where it is shown.

8] At the beginning of the Discussion (Line 5 page 6): Is "b" defined somewhere?

Typo mistakes:

9] Line 4-5 page 3 "the stress [...] are transferred"

10] Line 29 page 7 "the we find"

11] Line 2 page 8 "is and intuitive results"

---

## Author Comment (AC1) · 8 Sep 2016

RC - Referee Comment

AR - Authors' Response

RC. It stroke me at the first read, how much the context and references to previous or related works were lacking. There are few references to recent works related to sandpiles and other similar (lattice, mesoscopic) models. I think the authors should not limit their readings to models that explicit mention sandpiles or OFC in their titles, as there is a whole family of other models which share many interesting properties, and are morally very similar, if not exactly mapped to some particular variants of sandpiles.

In this respect, I am not completely sure that the results presented here are completely novel. They are probably novel to some extent, but should be compared with existing, close-by literature.

Overall, the paper is not badly written, but there are important things that are not easy to grasp, like the precise definition of how things were fitted or "calibrated", and the definitions of the quantities plotted in the figures. I detail what I mean by this in the points below.

AR. We thank the referee for his critical review that helped us in clarifying the context and novelty of the work. Our focus on the early works on the sandpile model, upon which our model is based, resulted in a limited context, i.e. how our model is situated relative to other discrete modeling approaches. In the following, we hope to address the points that he raised to clarify the results and to further highlight the novelty of the work.

RC1.0. About context: The following forest-fire model should, I think, be put in perspective with your work, as it shares some of its ingredients: "Forest-Fire Analogy to Explain the b Value of the Gutenberg-Richter Law for Earthquakes". More recently, the same author (and others, see references inside or citing it), did a paper dealing precisely with how the loading protocol affects the statistics of events in plasticity/fracture: "Avalanche-size distributions in mean-field plastic yielding models". That one is less tightly connected to your work, but it compares two kinds of driving (random and uniform), and it belongs to the family of models using "extremal dynamics" (loading all sites until exactly one is triggered).

In a sense, your protocol is quite similar to doing: - uniform loading at most steps. - with small proba p, trigger the most susceptible site by adding to it the needed stress (and maybe add the a fraction of that amount of stress to every other site). (in extremal dynamics, one adds the same amount of stress to all sites) I think it would actually increase the impact of your paper to connect your protocol and results to these other

existing protocols and results. You may want to check out, also, and see for yourself how relevant the following papers are: -"Universality in Sandpiles, Interface Depinning, and Earthquake Models", Paczuski et al (1996). -"Avalanche size distributions in mean field plastic yielding models" - (about SOC, 1 example:) "Dynamic scaling in stick-slip friction" (2005) - "Strain localization and anisotropic correlations in a mesoscopic model of amorphous plasticity", where "extremal dynamics" is used (there are countless others, not every author focuses on the dynamics and uses these words).

AR1.0. While we acknowledge the need for comparison with results from various classes of similar models, we would like to further clarify the protocol used in the paper, which is quite different from what the referee mentioned in his review.

Our protocol is based on the original rules of the sandpile model. Specifically, the external driving or growth is limited to one specific site that is randomly chosen. Unlike the OFC-based classes of models, our protocol does not involve uniform loading at all other sites. This main difference clearly sets us apart from most of the papers he recommended, majority of which require the uniform loading for all the sites.

We believe that, with this clarification on the nature of the protocol used, we have further clarified the novelty of the paper. Still, the references he mentioned have helped us to establish the similarities and differences of our work with others.

ACTION1.0. Wherever possible, we have placed additional references to put the paper into context. Most of the papers we added are the ones suggested by the referee.

Aside from the context-building, other remarks: RC1.1. Several times, you mention the similarity with the original sandpile model as a good thing, in particular the fact that the model is SOC seems to be very positive. For instance in the introduction, you say: "make the model more truthful to the original sandpile design, presenting a clear association with seismicity and SOC.". Why do you consider SOC and sandpiles to be a good thing by itself?

As Fisher noted in his 1998 Review: "Whether critical behavior is considered "selforganized" or not is somewhat a matter of taste: if the systems we are considering are driven at very slow velocity, then they will be very close to critical. In another well known situation, when a fluid is stirred on large scales, turbulence exists on a wide range of length scales extending down to the scale at which viscous dissipation occurs. In both of these and in many other contexts the parameter which is "tuned" to get a wide range of scales is the ratio of some basic "microscopic" scale to the scale at which the system is driven."

I don't think I could explain better than him: SOC is nowadays, to many people, not a particularly relevant characteristic. Often, what is called SOC is just a model where the critical point is at 0 or infinity. In your case, it is the system size which acts as a limiting size for the avalanche (dissipation occurs only at the boundaries in your model, not in the bulk, if I understood well.) By the way, your driving depends on the system's state, which in a sense can be seen as a feedback loop... thus giving a weaker "self-"organized structure to the problem.

I think this perspective on SOC and sanpiles should be updated/deleted, as they do not seem to bring anything to the paper. If you persist in liking SOC so much, you should give some explanation of why it is such a good thing that your model has some SOC in it.

AR1.1. The point being raised by the referee is a valid one, to which we also agree. In fact, for this work, the SOC idea has no particular usefulness, as far as the results are concerned. What we would like to highlight, which has not been communicated clearly in our current paper, is the fact that our paper has introduced minimal changes into the original sandpile model, which is the paradigm model of self-organized criticality. That the model is close to the original sandpile rules and may therefore retain SOC characteristics (although, the referee's point on the feedback is also valid) is just one of our results.

Why, then, a particular emphasis on the closeness to the sandpile model? Apart from the fact that it is one of the earliest discrete models of complexity, the original sandpile model is not able to capture the space and time characteristics of seismicity. Rules based on single triggering at random directions will result in normal (Gaussian) distributions of inter-event distances and times, which is not observed for seismicity. This, we believe, is the reason why subsequent models had to introduce uniform loading at all sites (see RC1.0 and AR1.0), along with asymmetry in the redistribution rules and dissipation in some cases. Here, we introduced a simple bias for a fraction of triggering times (and this fraction is not large, with around 10-3-10-2 recovering similar statistics as the data) and recovered both the interevent distances and time distributions, along with the magnitude (energy) distributions. The simplicity of the change introduced on the original sandpile and its corresponding recovery of the spatiotemporal statistics is one of the strengths of our paper.

ACTION1.1. We revised the paper accordingly to be able to emphasize our motivation and the novelty of our work. The Abstract and the Introduction now removes any mention of SOC; instead, we highlighted the fact that the original sandpile is unable to account for such observations, and explained how our model, with minimal parameters introduced, was able to recover similar statistical features of seismicity.

RC1.2. p.3, line 8: You say: "The number of affected sites in the grid, A, is used as a proxy for the actual energy ...". I think you should count the number of activations, not the number of sites activated once or more. If some sites are activated twice (or more), they should count twice (or more). Given that you put no dissipation, I guess it can occur ... maybe often? If this is indeed what you measured, be more clear. If you did not, you should show both quantities. The number of sites activated (irrespective of the number of activations) represents the area of EQs. The number of activations represents the seismic moment (energy released).

AR1.2. The choice of presenting the number of sites activated (the area A) is made to further strengthen the similarity of the model with the simple sandpile. To us, this

simplicity in the model dynamics is still the most important feature of the work.

We understand, however, the referee's concern. Apart from A, many other parameters can be used for characterizing the magnitude or scale of an event. For example, one can use the actual stress value (which we can call S) that was distributed among the neighborhoods during an avalanche event. We have used this metric in a previous work on landslides (Juanico, et al. Geophys. Res. Lett. 35, L19403, 2008), and we have observed that it scales as S âĹİ A3/2.

Here, the referee proposes that we count the number of actual activations (we can call this V). We believe that the motivation for tracking V is the fact that it may be closer to the actual dynamics of energy release during an earthquake event. In the following Figure 1, we show the probability density functions Prob(V) and the behavior of V(A).

Figure 1. (a) Prob(V) shows a similar behavior as Prob(A) [not shown here but present in the manuscript], i.e. it is quite robust to variations in p. (b) The V(A) plot shows a scaling behavior V âĹİ A3/2 for p = 0; at the extreme case of p = 1, the scattergrams show dual scaling, with a second scaling V âĹİ A4/3.

The probability distributions of V behave similarly as those of A (see manuscript) in their robustness to the introduction of p. The obtained scaling exponents are slightly different from those of p(A), however, due to the nonlinear scaling of V(A), as shown in Figure i(b). Near the p = 0 case, the scaling behavior is around VâĹİA3/2 suggesting that V is similar to S as a metric for the energy or volume. In the extreme case of p = 1, the scaling changes to approximately VâĹİA4/3, which can be easily explained by the nature of p; if p = 1, the most susceptible site will be targeted every time, which means that there will be minimal cases of reactivation, because the neighboring sites would always be depleted; the same area, therefore, will correspond to slightly lower volumes.

On the matter of correspondence: Because both the V and the A represent a relative measure of the extent of the relaxation of the system, and in fact show a scaling rela-

tion, we believe that both are equally valid representations of the energy being released in an earthquake.

ACTION1.2. We included the above discussion in the revisions.

RC1.3. same place: is used as a proxy for the actual energy or magnitude of replace "magnitude" with "seismic moment", as only this quantity has the dimension of an energy. Magnitude is the log (up to a prefactor) of the seismic moment.

AR1.3. We thank the referee for this correction.

ACTION1.3. We have revised the statement according to the referee's suggestion.

RC1.4. A general, important criticism: it is not clear how much you calibrated to get results to fit experimental data. More precise, yet clear explanation/discussion of the number of degrees of freedom (fitting parameters) in your fits, would be welcomed. Otherwise, the whole point of the paper (i.e. that your fits are rather good and relying on few fitting parameters) is compromised.

In this respect, I find the figures not very clear. In general, the methods should be clearly explained. Using a few more explicit sub-titles along your presentation may help.

AR1.4. This point, which has also been raised by the other referee (see RC2.6), is an important one that we would like to address in our revised paper. Upon reviewing our results, we realized that the term "calibration" might be a bit of a stretch. In fact, for the most part, our paper has presented analogies and similarities, and, although we believe that there is a correspondence between our model parameters and the actual conditions on the ground, we did not attempt to find such an exact relation.

As such, we concur with the second referee's opinion that the procedure we conducted is a simple rescaling of our model results for visual comparison with the empirical distributions. This change in terminology and perspective, we believe, does not diminish the value of our model results. The fact that such a comparison is even possible with

just a simple multiplication by a scalar is a testament to the feasibility of the model for capturing the features of the seismicity. Moreover, the rescaling parameters that we used may be explained using actual physical bases, and not obtained arbitrarily. The origin of the rescaling factors is better explained in the revised paper.

ACTION 1.4. We revised the paper to remove any mention of "calibration" and to instead reflect this change in perspective. The origin of the scaling factors for both R and T has also been explained in detail.

RC1.5. Related to my point (1): discuss also maybe, how often (what proba) is it that the site triggered was the most susceptible site of the state after the previous avalanche? By this I mean, if you record the position of the most loaded site after an event, how often is it that the triggering site of the next event is precisely the same site? I suspect this is much larger than p. I think it is good to discuss this, as it is a natural question the reader may have, and it could help relate your model to others.

AR1.5. We thank the reviewer for this helpful insight. As expected, due to the random nature of the triggering, in some instances, the most susceptible site will be targeted even without the action of the targeted triggering probability.

In Figure 2 below, we present the results of sample runs for different p values (grid dimension $L = 256$, iteration time $T = 107$), wherein the "natural" triggering of the most susceptible site (i.e. without the action of p) is tracked alongside all the instances of such triggering (i.e. including the targeted cases). The natural triggering is found to be hovering about its expected value, which is $[1/L^2]T, = 153$ where $1/L^2$ is the random chance of the most susceptible site to be targeted in the grid. The effect of the targeted triggering probability p is found to be order of magnitudes greater than this baseline value.

Figure 2. (red) Baseline values of the natural triggering of the most susceptible site for $p = 0$ compared with (blue) the total triggering for nonzero values of p.

ACTION1.5. The above discussion is incorporated in the revised paper.

RC1.6. By the way, in your model, is the area VS energy scaling in some way? These features are expected to have multiple scaling behavior, is it the case in your model? I recently published a study of various models, discussing this particular observable as a benchmark of model's quality, i.e. the area-magnitude scaling relationship: "Scaling laws in earthquake occurrence: Disorder, viscosity, and finite size effects in OlamiFeder-Christensen models" This can be a tricky thing to study, and the fact you do not recover expected natural data behaviour for this observable does not discard your model as uninteresting. I am just suggesting this as possible directions for future work.

AR1.6. (See also AR1.2) In Figure i(b), we show the scatter plots of A vs. V. Although we have not investigated the scaling behavior of these two parameters in detail, the plots show that the p = 0 case (original sandpile) follows a single scaling function VâĹİA3/2. On the other hand, the other extreme case of p = 1 shows an asymptotic behavior towards VâĹİA4/3. Visual inspection of p = 1, however, appears to show that the smallest A values follow the VâĹİA3/2 trend, up to a certain value. This preliminary analysis, however, may need to be checked more thoroughly for various p values.

ACTION1.6. As this result may need additional analyses, we leave out the discussion of this result in the revised paper.

RC1.7. figures: do not write PDF, but rather P(A), P(E), P(T), etc. (or Prob(A), etc, as you prefer). It would be more clear.

AR1.7. The original intention was to not use P(A), etc., to avoid any possible confusion of P with the parameter p in the model. But we agree with the referee that the use of a generic "PDF" label is confusing. In this case, we followed the referee's suggestion and used Prob(A), etc., wherever applicable.

ACTION 1.7. All figures that show a probability density plot now has y-axis labels of

Prob(...).

RC1.8. "... observed in the generation of earthquakes, which, despite regional differences, produce universal GR distributions.". This statement is rather controversial, and should be supported at least by a citation. One should not be overly confident with Per Bak's statements, (which were overly enthusiastic about SOC... and sometimes plainly wrong).

Geophysicists are quite interested in knowing the GR law region by region. It has regional differences, and integrating over all regions does not necessarily carry lots of physical meaning.

AR1.8. We acknowledge that the statement may be quite controversial and does not represent the prevailing consensus among researchers in the field. We thank the referee for this comment.

ACTION1.8. We have revised the statement to properly put the result in context.

RC1.9. "However, for the threshold Ath=50 used, we have not seen the power-law regimes due to the ..." I did not understood the definition of quantity $A_th. Is it given somewhere? If not, give it. If so, make it more visible.$

AR1.9. The point being made here is the fact that in the model, all the avalanches can be recorded down to the smallest possible ones. In contrast, for seismicity, there is a limit to our capability to record the smallest earthquake events; apart from the fact that they are too weak for accurate identification, the GR law predicts that their occurrence will be orders of magnitude greater than the lower-magnitude events. To mimic this limit in the empirical data, we introduced a threshold magnitude Ath, wherein avalanche events A < Ath are removed in the series.

ACTION1.9. We added a discussion of the Ath and the motivation for their use.

RC1.10. "In Figure 2(b)-(d), the we find that the rescaled model statistics for p=0.007 show good agreement". Correct the typo.

AR1.10. We have corrected the typo as noted by the referee.

ACTION 1.10: The text now reads: "In Figure 2(b)-(d), we find that the. . ." [see also RC2.10]

Please also note the supplement to this comment:
http://www.nonlin-processes-geophys-discuss.net/npg-2016-28/npg-2016-28-AC1-supplement.pdf

[Figure]

**Fig. 1.**

[Figure]

**Fig. 2.**

[Figure]

---

## Author Comment (AC2) · 8 Sep 2016

RC - Referee's Comments

AR - Authors' Response

RC. I think that overall the article is well written and the presented results are sufficiently clear, even though some passages are poor of references and some of the methods that were used are not fully explained.

AR. We thank the referee for his/her critical comments that helped improve the readability and further highlighted our important results. In the following, we offer a point-by-point response to his/her concerns.

[Figure]

In more detail:

RC2.1. The authors refer to the Zhang sandpile, introducing a deterministic toppling rule (the stress from the toppling site is equally redistributed between its nearest neighbors). It is known that the deterministic sandpile exhibits anomalous multi-scaling because of the breakdown of ergodicity caused by the existence of many toppling invariants [Bagnoli et al. 2003 Europhys. Lett. 63 512]. (If it isn't too annoying) I would suggest to choose a random toppling rule instead (that guarantees the model to belong to Manna universality class).

AR2.1. The referee offers a useful suggestion that we may consider for a future work. In the paper, however, one of our main arguments about the novelty of the work is the fact that it has captured the spatiotemporal signatures of seismicity while introducing very small tweaks in the sandpile model, which does not produce the same spatiotemporal signatures (see also RC1.1, AR1.1). As such, we opted to focus instead on the current model presented, while pursuing further investigations along the referee's suggested direction.

ACTION2.1. A short discussion on the sandpile model's inability to reproduce the statistical signatures of seismicity is added in the Introduction of the paper.

RC2.2. Why is the exponent of the avalanche size probability density functions ($\alpha=1.6$) so different from the exponent found for 2-dimensional sandpiles under synchronous updating rule ($\alpha=1.26$) -even in the limit of p=0- ? I think that this should be discussed or some literature should be cited. The only cited article with respect to this issue is Paguirigan et al., 2015, which deals with the introduction of sinks leading to non-conservation. I would suggest to clarify how does this relate to asynchronous updating rule. The other cited article (Lubeck, 1997) compares the static and dynamical properties of Zhang sandpile (the same that is considered here) with those of the Abelian sandpile model of Bak, Tang, and Wiesenfeld, stating that the exponents of the avalanche probability distribution are the same. I think that some more

(and more relevant) references should be cited here, in order to give stronger evidence for the appeareance of the exponent 1.6 (given that it appears to fit really well with the experimental data).

AR2.2. The referee raises a valid point, which we have also noted. In the revised paper, we added a short discussion on the possible reason for the occurrence of slightly steeper power-law distributions.

The asynchronicity, which, in here, is defined as the addition of a single trigger value to a randomly chosen (or, in this case, preferred) site, tends to produce isolated regions that are near the threshold value. Because of the lack of global connectivity among such sites (which would have been possible if synchronous updating were used), we observe the preponderance of small avalanche events that affect only small local neighborhoods at a time. This, in turn, leads to a corresponding decrease in the occurrence of very large-area avalanches.

ACTION2.2. The above discussion is included in the revised text.

RC2.4. In Figure 2 (b)-(d) I guess that black dots represent shuffled data but I think that it should be written explicitly for better readability.

AR2.4. We thank the reviewer for pointing out this oversight. The insets in Figure 2 (b)-(d) correspond to shuffled sequences for both the data (colored markers) and the model (black dots), to provide a comparison between the statistical distribution of the data (i.e. earthquake sequences with memory) and a randomized sequence (i.e. no memory), similar to the work done by Batac and Kantz [NPG 2014]. This also leads to the crossover value R* where the data and the shuffled sequences begin to show similar trends; this, in turn, will be used for creating the conditional distributions $T_{in} = \{T \mid R \leq R^*\}$ and $T_{out} = \{T \mid R > R^*\}$.

ACTION2.4. In the revised paper, we emphasized this procedure by removing the insets of the original Figure 2 and plotting the same in a new figure, for clarity.

RC2.5. The authors state and show in Fig. 2 that the distribution of the spatial distances between events becomes bimodal for some values of p, stating that "bimodality [in the experimentally observed distributions] is due to the difference in the characteristic times of the correlated aftershock sequences and the independent mainshocks". I think it would be useful that the authors discuss why such distribution becomes bimodal in the model, for some intermediate values of p.

AR2.5. The p describes the persistence of the system to target the most susceptible site in the grid, which is most often lies at the vicinity of a previous avalanche. Consequently, this gives preponderance for small spatial separations, thereby producing a bimodal pattern in the trend, similar to what is observed empirically. Unlike previous implementations, however, the p is just a stochastic term, and does not "force" the production of small-distances, by, say, pre-selecting the region to perturb. In essence, we have utilized a property of the sandpile grid; by only imposing that the targeted triggering be at the most susceptible site, we have obtained short characteristic distances because the most susceptible site is oftentimes within the vicinity of a previous collapse.

ACTION2.5. The above discussion is introduced in the revised paper.

RC2.6. I think that the "calibration with real-world data" procedure is not perfectly clear. I would suggest to explain the procedure in more detail. As it is, I can imagine it is only a way to plot simulations results and experimental data on the same plot, in other world it is an artifact that allows to compare quatitatively two data sets that in principle can only be compared qualitatively but that does not explicitly add any new information to the article.

AR2.6. This point has also been raised by the other referee (see RC1.4), and represents a key idea that needed further elucidation in our paper.

We thank the referee for pointing out that the procedure we conducted is indeed a simple rescaling of the model results for a better visual comparison with the data. How

does this change in perspective affect the importance of the results? We would like to think that the model results are still important and novel. If the model results were not in correspondence with those of the empirical data distributions, then we would expect complicated rescaling functions to "match" the model with data. However, for all our results, we find that the model results easily correspond with the empirical data with just a simple multiplication by a scalar, which, in effect, is similar to just converting one cell unit into actual physical length and one iteration into a unit of time. Finally, we found that the scaling factors used for making the visual correspondence can be derived from a physical basis, and not arbitrarily obtained.

ACTION2.6. We provided a change in perspective and removed any mention of "calibration" in the revised text. Instead, we present the results as simple visual comparisons. The choice of the scaling factors used has also been explained in detail.

RC2.7. In Line 3 page 6 the authors write: "The GR law [...] can be shown to be equivalent to an energy E CCDF ...". I think it would be useful to cite here where it is shown.

AR2.7. For this purpose, we use the discussion given in the Introduction of the paper by Jagla [Phys. Rev. Lett. 111, 238501, 2013] to explain how the exponents are derived.

ACTION2.7. The above reference is cited in the mentioned discussion.

RC2.8. At the beginning of the Discussion (Line 5 page 6): Is "b" defined somewhere?

AR2.8. The GR law is usually given in terms of magnitudes M and introduced in the complimentary cumulative distribution (CCDF) form given by $\log\_10âĄąN=a-bM$, where b gives a constant value close to 1 for regions that are seismically active.

ACTION2.8. The above discussion is added in the text.

Typo mistakes:

RC2.9) Line 4-5 page 3 "the stress [...] are transferred" RC2.10) Line 29 page 7 "the

we find" RC2.11) Line 2 page 8 "is and intuitive results"

AR2.9-11. We have corrected the specified typo errors.

ACTION 2.9: The text now reads: "the stress [. . .] is transferred. . ." ACTION 2.10. The text now reads: "In Figure 2(b)-(d), we find that the. . ." [see also RC1.10] ACTION 2.11. The text now reads: "is an intuitive result. . ."

Please also note the supplement to this comment:
http://www.nonlin-processes-geophys-discuss.net/npg-2016-28/npg-2016-28-AC2-supplement.pdf

**Supplement:**

**Sandpile-based model for capturing magnitude distributions and spatiotemporal clustering and separation in regional earthquakes**

R. C. Batac, A. A. Paguirigan Jr., A. B. Tarun, and A. G. Longjas

RC – Referee Comments
AR – Author Response

**Response to Comments: Referee #1 François P. Landes** (francois.landes@gmail.com)

It stroke me at the first read, how much the context and references to previous or related works were lacking. There are few references to recent works related to sandpiles and other similar (lattice, mesoscopic) models. I think the authors should not limit their readings to models that explicit mention sandpiles or OFC in their titles, as there is a whole family of other models which share many interesting properties, and are morally very similar, if not exactly mapped to some particular variants of sandpiles. In this respect, I am not completely sure that the results presented here are completely novel. They are probably novel to some extent, but should be compared with existing, close-by literature.

Overall, the paper is not badly written, but there are important things that are not easy to grasp, like the precise definition of how things were fitted or "calibrated", and the definitions of the quantities plotted in the figures. I detail what I mean by this in the points below.

We thank the referee for his critical review that helped us in clarifying the context and novelty of the work. Our focus on the early works on the sandpile model, upon which our model is based, resulted in a limited context, i.e. how our model is situated relative to other discrete modeling approaches. In the following, we hope to address the points that he raised to clarify the results and to further highlight the novelty of the work.

**RC1.0. About context:**

The following forest-fire model should, I think, be put in perspective with your work, as it shares some of its ingredients: "Forest-Fire Analogy to Explain the b Value of the Gutenberg-Richter Law for Earthquakes". More recently, the same author (and others, see references inside or citing it), did a paper dealing precisely with how the loading protocol affects the statistics of events in plasticity/fracture: "Avalanche-size distributions in mean-field plastic yielding models". That one is less tightly connected to your work, but it compares two kinds of driving (random and uniform), and it belongs to the family of models using "extremal dynamics" (loading all sites until exactly one is triggered).

In a sense, your protocol is quite similar to doing:
- uniform loading at most steps.
- with small proba p, trigger the most susceptible site by adding to it the needed stress (and maybe add the a fraction of that amount of stress to every other site). (in extremal dynamics, one adds the same amount of stress to all sites) I think it would actually increase the impact of your paper to connect your protocol and results to these other existing protocols and results. You may want to check out, also, and see for yourself how relevant the following papers are:
-"Universality in Sandpiles, Interface Depinning, and Earthquake Models", Paczuski et al (1996).

-"Avalanche size distributions in mean field plastic yielding models"
- (about SOC, 1 example:) "Dynamic scaling in stick-slip friction" (2005)
- "Strain localization and anisotropic correlations in a mesoscopic model of amorphous plasticity", where "extremal dynamics" is used (there are countless others, not every author focuses on the dynamics and uses these words).

**AR1.0.** While we acknowledge the need for comparison with results from various classes of similar models, we would like to further clarify the protocol used in the paper, which is quite different from what the referee mentioned in his review.

Our protocol is based on the original rules of the sandpile model. Specifically, the external driving or growth is limited to *one specific site* that is randomly chosen. Unlike the OFC-based classes of models, our protocol *does not involve uniform loading* at all other sites. This main difference clearly sets us apart from most of the papers he recommended, majority of which require the uniform loading for all the sites.

We believe that, with this clarification on the nature of the protocol used, we have further clarified the novelty of the paper. Still, the references he mentioned have helped us to establish the similarities and differences of our work with others.

ACTION1.0. Wherever possible, we have placed additional references to put the paper into context. Most of the papers we added are the ones suggested by the referee.

Aside from the context-building, other remarks:
**RC1.1.** Several times, you mention the similarity with the original sandpile model as a good thing, in particular the fact that the model is SOC seems to be very positive. For instance in the introduction, you say: "make the model more truthful to the original sandpile design, presenting a clear association with seismicity and SOC.". Why do you consider SOC and sandpiles to be a good thing by itself?

As Fisher noted in his 1998 Review: "Whether critical behavior is considered "selforganized" or not is somewhat a matter of taste: if the systems we are considering are driven at very slow velocity, then they will be very close to critical. In another well known situation, when a fluid is stirred on large scales, turbulence exists on a wide range of length scales extending down to the scale at which viscous dissipation occurs. In both of these and in many other contexts the parameter which is "tuned" to get a wide range of scales is the ratio of some basic "microscopic" scale to the scale at which the system is driven."

I don't think I could explain better than him: SOC is nowadays, to many people, not a particularly relevant characteristic. Often, what is called SOC is just a model where the critical point is at 0 or infinity. In your case, it is the system size which acts as a limiting size for the avalanche (dissipation occurs only at the boundaries in your model, not in the bulk, if I understood well.) By the way, your driving depends on the system's state, which in a sense can be seen as a feedback loop... thus giving a weaker "self-"organized structure to the problem.

I think this perspective on SOC and sanpiles should be updated/deleted, as they do not seem to bring anything to the paper. If you persist in liking SOC so much, you should give some explanation of why it is such a good thing that your model has some SOC in it.

**AR1.1.** The point being raised by the referee is a valid one, to which we also agree. In fact, for this work, the SOC idea has no particular usefulness, as far as the results are concerned. What we would like to highlight, which has not been communicated clearly in our current paper, is the fact that our paper has introduced minimal changes into the original sandpile model, which is the paradigm model of self-organized criticality. That the model is close to the original sandpile rules and may therefore retain SOC characteristics (although, the referee's point on the feedback is also valid) is just one of our results.

Why, then, a particular emphasis on the closeness to the sandpile model? Apart from the fact that it is one of the earliest discrete models of complexity, the original sandpile model is not able to capture the space and time characteristics of seismicity. Rules based on single triggering at random directions will result in normal (Gaussian) distributions of inter-event distances and times, which is not observed for seismicity. This, we believe, is the reason why subsequent models had to introduce uniform loading at all sites (see **RC1.0** and **AR1.0**), along with asymmetry in the redistribution rules and dissipation in some cases. Here, we introduced a simple bias for a fraction of triggering times (and this fraction is not large, with around $10^{-3}$-$10^{-2}$ recovering similar statistics as the data) and recovered both the interevent distances and time distributions, along with the magnitude (energy) distributions. The simplicity of the change introduced on the original sandpile and its corresponding recovery of the spatiotemporal statistics is one of the strengths of our paper.

ACTION1.1. We revised the paper accordingly to be able to emphasize our motivation and the novelty of our work. The Abstract and the Introduction now removes any mention of SOC; instead, we highlighted the fact that the original sandpile is unable to account for such observations, and explained how our model, with minimal parameters introduced, was able to recover similar statistical features of seismicity.

**RC1.2. p.3, line 8:**
You say: "The number of affected sites in the grid, A, is used as a proxy for the actual energy …". I think you should count the number of activations, not the number of sites activated once or more. If some sites are activated twice (or more), they should count twice (or more). Given that you put no dissipation, I guess it can occur … maybe often? If this is indeed what you measured, be more clear. If you did not, you should show both quantities. The number of sites activated (irrespective of the number of activations) represents the area of EQs. The number of activations represents the seismic moment (energy released).

**AR1.2.** The choice of presenting the number of sites activated (the area $A$) is made to further strengthen the similarity of the model with the simple sandpile. To us, this simplicity in the model dynamics is still the most important feature of the work.

We understand, however, the referee's concern. Apart from $A$, many other parameters can be used for characterizing the magnitude or scale of an event. For example, one can use the actual

stress value (which we can call $S$) that was distributed among the neighborhoods during an avalanche event. We have used this metric in a previous work on landslides (Juanico, et al. Geophys. Res. Lett. 35, L19403, 2008), and we have observed that it scales as $S \propto A^{3/2}$.

Here, the referee proposes that we count the number of actual activations (we can call this $V$). We believe that the motivation for tracking $V$ is the fact that it may be closer to the actual dynamics of energy release during an earthquake event. In the following Figure i, we show the probability density functions Prob($V$) and the behavior of $V(A)$.

[Figure]

Figure i. (a) Prob($V$) shows a similar behavior as Prob($A$) [not shown here but present in the manuscript], i.e. it is quite robust to variations in $p$. (b) The $V(A)$ plot shows a scaling behavior $V \propto A^{3/2}$ for $p = 0$; at the extreme case of $p = 1$, the scattergrams show dual scaling, with a second scaling $V \propto A^{4/3}$.

The probability distributions of $V$ behave similarly as those of $A$ (see manuscript) in their robustness to the introduction of $p$. The obtained scaling exponents are slightly different from those of $p(A)$, however, due to the nonlinear scaling of $V(A)$, as shown in Figure i(b). Near the $p = 0$ case, the scaling behavior is around $V \propto A^{3/2}$ suggesting that $V$ is similar to $S$ as a metric for the energy or volume. In the extreme case of $p = 1$, the scaling changes to approximately $V \propto A^{4/3}$, which can be easily explained by the nature of $p$; if $p = 1$, the most susceptible site will be targeted every time, which means that there will be minimal cases of reactivation, because the neighboring sites would always be depleted; the same area, therefore, will correspond to slightly lower volumes.

On the matter of correspondence: Because both the $V$ and the $A$ represent a relative measure of the extent of the relaxation of the system, and in fact show a scaling relation, we believe that both are equally valid representations of the energy being released in an earthquake.

ACTION1.2. We included the above discussion in the revisions.

**RC1.3. same place:** is used as a proxy for the actual energy or magnitude of replace "magnitude" with "seismic moment", as only this quantity has the dimension of an energy. Magnitude is the log (up to a prefactor) of the seismic moment.

**AR1.3.** We thank the referee for this correction.

ACTION1.3. We have revised the statement according to the referee's suggestion.

**RC1.4.** A general, important criticism: it is not clear how much you calibrated to get results to fit experimental data. More precise, yet clear explanation/discussion of the number of degrees of freedom (fitting parameters) in your fits, would be welcomed. Otherwise, the whole point of the paper (i.e. that your fits are rather good and relying on few fitting parameters) is compromised.

In this respect, I find the figures not very clear. In general, the methods should be clearly explained. Using a few more explicit sub-titles along your presentation may help.

**AR1.4.** This point, which has also been raised by the other referee (see **RC2.6**), is an important one that we would like to address in our revised paper. Upon reviewing our results, we realized that the term "calibration" might be a bit of a stretch. In fact, for the most part, our paper has presented analogies and similarities, and, although we believe that there is a correspondence between our model parameters and the actual conditions on the ground, we did not attempt to find such an exact relation.

As such, we concur with the second referee's opinion that the procedure we conducted is a simple rescaling of our model results for visual comparison with the empirical distributions. This change in terminology and perspective, we believe, does not diminish the value of our model results. The fact that such a comparison is even possible with just a simple multiplication by a scalar is a testament to the feasibility of the model for capturing the features of the seismicity. Moreover, the rescaling parameters that we used may be explained using actual physical bases, and not obtained arbitrarily. The origin of the rescaling factors is better explained in the revised paper.

ACTION 1.4. We revised the paper to remove any mention of "calibration" and to instead reflect this change in perspective. The origin of the scaling factors for both $R$ and $T$ has also been explained in detail.

**RC1.5.** Related to my point (1): discuss also maybe, how often (what proba) is it that the site triggered was the most susceptible site of the state after the previous avalanche? By this I mean, if you record the position of the most loaded site after an event, how often is it that the triggering site of the next event is precisely the same site? I suspect this is much larger than p. I think it is good to discuss this, as it is a natural question the reader may have, and it could help relate your model to others.

**AR1.5.** We thank the reviewer for this helpful insight. As expected, due to the random nature of the triggering, in some instances, the most susceptible site will be targeted even without the action of the targeted triggering probability.

In Figure ii below, we present the results of sample runs for different $p$ values (grid dimension $L$ = 256, iteration time $T = 10^7$), wherein the "natural" triggering of the most susceptible site (i.e. without the action of $p$) is tracked alongside all the instances of such triggering (i.e. including the targeted cases). The natural triggering is found to be hovering about its expected value, which is $[1/L^2]T$, = 153 where $1/L^2$ is the random chance of the most susceptible site to be targeted in the grid. The effect of the targeted triggering probability $p$ is found to be order of magnitudes greater than this baseline value.

[Figure]

Figure ii. (red) Baseline values of the natural triggering of the most susceptible site for $p = 0$ compared with (blue) the total triggering for nonzero values of $p$.

ACTION1.5. The above discussion is incorporated in the revised paper.

**RC1.6.** By the way, in your model, is the area VS energy scaling in some way? These features are expected to have multiple scaling behavior, is it the case in your model? I recently published a study of various models, discussing this particular observable as a benchmark of model's quality,

i.e. the area-magnitude scaling relationship: "Scaling laws in earthquake occurrence: Disorder, viscosity, and finite size effects in OlamiFeder-Christensen models" This can be a tricky thing to study, and the fact you do not recover expected natural data behaviour for this observable does not discard your model as uninteresting. I am just suggesting this as possible directions for future work.

**AR1.6**. (See also **AR1.2**) In Figure i(b), we show the scatter plots of $A$ vs. $V$. Although we have not investigated the scaling behavior of these two parameters in detail, the plots show that the $p = 0$ case (original sandpile) follows a single scaling function $V \propto A^{3/2}$. On the other hand, the other extreme case of $p = 1$ shows an asymptotic behavior towards $V \propto A^{4/3}$. Visual inspection of $p = 1$, however, appears to show that the smallest $A$ values follow the $V \propto A^{3/2}$ trend, up to a certain value. This preliminary analysis, however, may need to be checked more thoroughly for various $p$ values.

ACTION1.6. As this result may need additional analyses, we leave out the discussion of this result in the revised paper.

**RC1.7.** figures: do not write PDF, but rather P(A), P(E), P(T), etc. (or Prob(A), etc, as you prefer). It would be more clear.

**AR1.7.** The original intention was to not use P(A), etc., to avoid any possible confusion of P with the parameter p in the model. But we agree with the referee that the use of a generic "PDF" label is confusing. In this case, we followed the referee's suggestion and used Prob(A), etc., wherever applicable.

ACTION 1.7. All figures that show a probability density plot now has y-axis labels of Prob(…).

**RC1.8.** "… observed in the generation of earthquakes, which, despite regional differences, produce universal GR distributions.". This statement is rather controversial, and should be supported at least by a citation. One should not be overly confident with Per Bak's statements, (which were overly enthusiastic about SOC… and sometimes plainly wrong).

Geophysicists are quite interested in knowing the GR law region by region. It has regional differences, and integrating over all regions does not necessarily carry lots of physical meaning.

**AR1.8.** We acknowledge that the statement may be quite controversial and does not represent the prevailing consensus among researchers in the field. We thank the referee for this comment.

ACTION1.8. We have revised the statement to properly put the result in context.

**RC1.9.** "However, for the threshold Ath=50 used, we have not seen the power-law regimes due to the …" I did not understood the definition of quantity A_th. Is it given somewhere? If not, give it. If so, make it more visible.

**AR1.9.** The point being made here is the fact that in the model, all the avalanches can be recorded down to the smallest possible ones. In contrast, for seismicity, there is a limit to our capability to

record the smallest earthquake events; apart from the fact that they are too weak for accurate identification, the GR law predicts that their occurrence will be orders of magnitude greater than the lower-magnitude events.  To mimic this limit in the empirical data, we introduced a threshold magnitude $A_{th}$, wherein avalanche events $A < A_{th}$ are removed in the series.

ACTION1.9. We added a discussion of the $A_{th}$ and the motivation for their use.

**RC1.10.** "In Figure 2(b)-(d), the we find that the rescaled model statistics for p=0.007 show good agreement". Correct the typo.

**AR1.10.** We have corrected the typo as noted by the referee.

ACTION 1.10: The text now reads: "In Figure 2(b)-(d), we find that the…" [see also **RC2.10**]

**Response to Comments: Referee #2 Anonymous**

I think that overall the article is well written and the presented results are sufficiently clear, even though some passages are poor of references and some of the methods that were used are not fully explained.

We thank the referee for his/her critical comments that helped improve the readability and further highlighted our important results. In the following, we offer a point-by-point response to his/her concerns.

In more detail:
**RC2.1.** The authors refer to the Zhang sandpile, introducing a deterministic toppling rule (the stress from the toppling site is equally redistributed between its nearest neighbors). It is known that the deterministic sandpile exhibits anomalous multi-scaling because of the breakdown of ergodicity caused by the existence of many toppling invariants [Bagnoli et al. 2003 Europhys. Lett. 63 512]. (If it isn't too annoying) I would suggest to choose a random toppling rule instead (that guarantees the model to belong to Manna universality class).

**AR2.1.** The referee offers a useful suggestion that we may consider for a future work. In the paper, however, one of our main arguments about the novelty of the work is the fact that it has captured the spatiotemporal signatures of seismicity while introducing very small tweaks in the sandpile model, which does not produce the same spatiotemporal signatures (see also **RC1.1**, **AR1.1**). As such, we opted to focus instead on the current model presented, while pursuing further investigations along the referee's suggested direction.

ACTION2.1. A short discussion on the sandpile model's inability to reproduce the statistical signatures of seismicity is added in the Introduction of the paper.

**RC2.2.** Why is the exponent of the avalanche size probability density functions ($\alpha=1.6$) so different from the exponent found for 2-dimensional sandpiles under synchronous updating rule ($\alpha=1.26$) -even in the limit of p=0- ? I think that this should be discussed or some literature should be cited. The only cited article with respect to this issue is Paguirigan et al., 2015, which deals with the introduction of sinks leading to non-conservation. I would suggest to clarify how does this relate to asynchronous updating rule. The other cited article (Lubeck, 1997) compares the static and dynamical properties of Zhang sandpile (the same that is considered here) with those of the Abelian sandpile model of Bak, Tang, and Wiesenfeld, stating that the exponents of the avalanche probability distribution are the same. I think that some more (and more relevant) references should be cited here, in order to give stronger evidence for the appearence of the exponent 1.6 (given that it appears to fit really well with the experimental data).

**AR2.2.** The referee raises a valid point, which we have also noted. In the revised paper, we added a short discussion on the possible reason for the occurrence of slightly steeper power-law distributions.

The asynchronicity, which, in here, is defined as the addition of a single trigger value to a randomly chosen (or, in this case, preferred) site, tends to produce isolated regions that are near the threshold value. Because of the lack of global connectivity among such sites (which would have been possible if synchronous updating were used), we observe the preponderance of small avalanche events that affect only small local neighborhoods at a time. This, in turn, leads to a corresponding decrease in the occurrence of very large-area avalanches.

ACTION2.2. The above discussion is included in the revised text.

**RC2.4.** In Figure 2 (b)-(d) I guess that black dots represent shuffled data but I think that it should be written explicitly for better readability.

**AR2.4.** We thank the reviewer for pointing out this oversight. The insets in Figure 2 (b)-(d) correspond to shuffled sequences for both the data (colored markers) and the model (black dots), to provide a comparison between the statistical distribution of the data (i.e. earthquake sequences with memory) and a randomized sequence (i.e. no memory), similar to the work done by Batac and Kantz [NPG 2014]. This also leads to the crossover value $R^*$ where the data and the shuffled sequences begin to show similar trends; this, in turn, will be used for creating the conditional distributions $T_{in} = \{T \mid R \leq R^*\}$ and $T_{out} = \{T \mid R > R^*\}$.

ACTION2.4. In the revised paper, we emphasized this procedure by removing the insets of the original Figure 2 and plotting the same in a new figure, for clarity.

**RC2.5.** The authors state and show in Fig. 2 that the distribution of the spatial distances between events becomes bimodal for some values of p, stating that "bimodality [in the experimentally observed distributions] is due to the difference in the characteristic times of the correlated aftershock sequences and the independent mainshocks". I think it would be useful that the authors discuss why such distribution becomes bimodal in the model, for some intermediate values of p.

**AR2.5.** The *p* describes the persistence of the system to target the most susceptible site in the grid, which is most often lies at the vicinity of a previous avalanche. Consequently, this gives preponderance for small spatial separations, thereby producing a bimodal pattern in the trend, similar to what is observed empirically. Unlike previous implementations, however, the *p* is just a stochastic term, and does not "force" the production of small-distances, by, say, pre-selecting the region to perturb. In essence, we have utilized a property of the sandpile grid; by only imposing that the targeted triggering be at the most susceptible site, we have obtained short characteristic distances *because* the most susceptible site is oftentimes within the vicinity of a previous collapse.

ACTION2.5. The above discussion is introduced in the revised paper.

**RC2.6.** I think that the "calibration with real-world data" procedure is not perfectly clear. I would suggest to explain the procedure in more detail. As it is, I can imagine it is only a way to plot simulations results and experimental data on the same plot, in other world it is an artifact that

allows to compare quatitatively two data sets that in principle can only be compared qualitatively but that does not explicitly add any new information to the article.

**AR2.6.** This point has also been raised by the other referee (see **RC1.4**), and represents a key idea that needed further elucidation in our paper.

We thank the referee for pointing out that the procedure we conducted is indeed a simple rescaling of the model results for a better visual comparison with the data. How does this change in perspective affect the importance of the results? We would like to think that the model results are still important and novel. If the model results were not in correspondence with those of the empirical data distributions, then we would expect complicated rescaling functions to "match" the model with data. However, for all our results, we find that the model results easily correspond with the empirical data with just a simple multiplication by a scalar, which, in effect, is similar to just converting one cell unit into actual physical length and one iteration into a unit of time. Finally, we found that the scaling factors used for making the visual correspondence can be derived from a physical basis, and not arbitrarily obtained.

ACTION2.6. We provided a change in perspective and removed any mention of "calibration" in the revised text. Instead, we present the results as simple visual comparisons. The choice of the scaling factors used has also been explained in detail.

**RC2.7.** In Line 3 page 6 the authors write: "The GR law [...] can be shown to be equivalent to an energy E CCDF ...". I think it would be useful to cite here where it is shown.

**AR2.7.** For this purpose, we use the discussion given in the Introduction of the paper by Jagla [Phys. Rev. Lett. 111, 238501, 2013] to explain how the exponents are derived.

ACTION2.7. The above reference is cited in the mentioned discussion.

**RC2.8.** At the beginning of the Discussion (Line 5 page 6): Is "b" defined somewhere?

**AR2.8.** The GR law is usually given in terms of magnitudes $M$ and introduced in the complimentary cumulative distribution (CCDF) form given by $\log_{10} N = a - bM$, where $b$ gives a constant value close to 1 for regions that are seismically active.

ACTION2.8. The above discussion is added in the text.

Typo mistakes:

**RC2.9)** Line 4-5 page 3 "the stress [...] are transferred"
**RC2.10)** Line 29 page 7 "the we find"
**RC2.11)** Line 2 page 8 "is and intuitive results"

**AR2.9-11.** We have corrected the specified typo errors.

ACTION 2.9: The text now reads: "the stress [...] is transferred..."

ACTION 2.10. The text now reads: "In Figure 2(b)-(d), we find that the…" [see also **RC1.10**]
ACTION 2.11. The text now reads: "is an intuitive result…"

---

## Author Response (AR2)

10 January 2017

**Dr. Richard Gloaguen**

Editor

*Nonlinear Processes in Geophysics*

Dear Dr. Gloaguen:

We are attaching herewith our revised manuscript npg-2016-028 titled **Sandpile-based model for capturing magnitude distributions and spatiotemporal clustering and separation in regional earthquakes**, along with our response to the referees' comments.

We carefully considered all the points raised by the referees, which, we believe, improved the general readability and presentation of our results. More importantly, one of our major change for this version is on the interpretation of the measures used for the avalanche event sizes. While still presenting the avalanche area distributions as was done in the previous works we have cited, we noted that the number of activations is a better reepresentation of the energy being released in the earthquake events.

For this round of reviews, we have attached our point-by-point responses to the referees' comments, a list of our major changes, and a detailed list of corrections from our previous version. We hope that the revised manuscipt will be better suited for possible publication in the journal *Nonlinear Processes in Geophysics*.

We thank you for your consideration.

Sincerely,

**Rene C. Batac**

**Antonino A. Paguirigan**

**Anjali B. Tarun**

**Anthony G. Longjas**

Manuscript Number: npg-2016-28

**Sandpile-based model for capturing magnitude distributions and spatiotemporal clustering and separation in regional earthquakes**

Rene C. Batac, Antonino A. Paguirigan Jr., Anjali B. Tarun, and Anthony G. Longjas

**Response to Referee Reports**

RC – Referee's Comments

AR – Authors' Response

**Referee #1: Dr. Francois Landes, francois.landes@gmail.com**

*I think the authors have substantially improved the paper. There are still a few points that need to be improved, but the overall quality is now much more satisfactory.*

We thank the referee for agreeing to review the paper again. His suggestions have significantly improved the quality and overall presentation of the paper, both in this version and the previous one. We have incorporated additional revisions to the paper to address his further comments below.

**RC1.0. General comment:**

*Please use floating point notation (e.g. 1.5 instead of 3/2), when you have no prediction or argument for the expoenents to take these simple rational values. Otherwise it is misleading, as usually when an exponent is \*exactly\*U equal to a ratinal number, there is (relatively) simple argument explaining that result. In your paper I believe most exponents are irrational numbers.*

**AR1.0.** We agree with the referee's comment, and revised the paper accordingly. We have, in fact, conducted additional work to check whether the "nice-looking" rational exponents will come out from theoretical origins. As most of these works are still being pursued, and will not be included in the paper, we will use the empirical fits reported in decimal form.

ACTION1.0. In the text and in the figures, the exponents are reported in decimal form instead of fractions.

**RC1.1. [In response to previous review item 1.0]**

*Ok, I see you added some litterature, I haven't compared in detail but reading the whole paper I found it more clear.*
*I see you cited Landes and Lippiello 2016, this is nice but not very necessary: please do not feel like you should cite me because I'm refereeing. Cite if you truly believe it is relevant.*
*I think you would gain readership by further putting things into context in a precise way, comparing your results quantitatively with other model's, but that may also be for a separate publication, it's your choice. Now the reader is not lost, I think.*

**AR1.1.** On this account, many of the additional papers cited for breadth have been suggested by the referee in the previous review, for which we are thankful. We agree that these citations have placed the work in a better context, while still focusing on the simplicity of the approach.

We have cited Landes and Lippiello (2016) because it introduced us to the different physical quantities associated with the area $A$ and activations $V$. Prior to the review, we only thought of these model metrics as two different ways of representing the relative magnitude of an event in the grid. Through this paper, though, we have seen that these two quantities may actually be associated with area and seismic moment, respectively. The citation, therefore, is important. In fact, we have revised our interpretation of the results based on this suggestion [see **RC1.4** and **AR1.4**].

We sincerely hope that the revised paper will be more understandable to a wider readership. We again thank the referee for his critical assessment.

**RC1.2. [In response to previous review item 1.1]** Ok, good.

**AR1.2.** We thank the referee for clarifying the association (or lack of it) with self-organized criticality (SOC) in the previous review item 1.1. In the revised paper, we preserve the discussion of the main motivation of our model, which is to be able to simulate the statistical distributions of earthquakes through a sandpile-based model.

**RC1.3. [In response to previous review item 1.2]**
Thank you for this nice discussion and adding these itneresting results.
You should add one or two tentative fits and their corresponding power-law exponents in Fig 4.

**AR1.3.** The new results are based on the referees suggestions. Figure 4(a) now includes a power-law $V^{-1.45}$ as a guide to the eye. We revised the paper as suggested.

ACTION1.3. The new Figure 4(a) is revised to include a power-law guide to the eye. The repeated use of the parameter $\alpha$, already used in Figure 1 to denote the Prob($A$) exponent, is also avoided; in Figure 4(b), the $V$ vs. $A$ scaling is now represented by the scaling exponent $\delta$.

**RC1.4. [In response to previous review item 1.2]** About your last comment on 1.2, a remark:
you say that V and A are equally representations of the energy .... but it's like saying that velocity v and kinetic energy $E\_k$ of a system are equally valid representations of its temperature: instead, we have $|v|^2 \sim T \sim E\_k$, not $|v| \sim T \sim E\_k$.
Exponents change if you use a variable other than the correct one (or not proportional to it).
If $P(A) \sim A^{-1.6}$ and $V \sim A^{1.5}$ for instance, using $P(A) dA = P(V) dV$, you get (I think) $P(V) \sim V^{-((1.6-1.5+1)/1.5)} = V^{-1.4}$.
I do not think it is crucial for your results that the exponents match very well: the key result is that you have bimodal statistics in both space and time distributions which appear as a result of introducing p. So even if the exponent of fig 4 is not very close to the famous 5/3=1.6666666 you wish for, it's ok, your paper is worthwhile (amyway the "true" value of b is very debated).

**AR1.4.** This comment, along with the analogy presented, made us appreciate the difference between the two measures we have presented [see also **RC2.2** and **AR2.2**]. Interestingly, the Prob($V$) plots appear to follow the trend of $V^{-\beta}$,

where $\beta$ is between 1.4 to 1.5, as predicted by the referee from the scaling arguments above; we have presented the $\beta$=1.45 in Figure 4(a) as a guide to the eye.

Therefore, in the revised paper, we stick with the presentation of Figure 1, where the earthquake energy distributions are placed side by side with Prob($A$). This is in keeping with the earlier works on sandpile-based approaches, most of which have tracked only the area $A$. For Figure 4, on the other hand, a more mechanistic measure of the actual energy released is obtained through $V$. As the referee noted, the slight differences in exponents are not an issue; the goal is not to completely replicate the exponents. But it is worth noting that $\beta$ is still close to the obtained empirical exponent.

ACTION1.4. The above-mentioned points are highlighted in the revised paper.

**RC1.5. [In response to previous review item 1.3]**
Ok... So you elected to call M (and sometimes m) the magnitude and m (?) its threshold... why not use m always and m_th for the threshold ?

**AR1.5.** We did as the referee suggested.

ACTION1.5. We used $m_{th}$ to denote the threshold magnitude and $m$ for the magnitude in the revised paper.

**RC1.6. [In response to previous review item 1.4]**
Ok, excellent, this point is now much clearer to me and clearer in the paper.

**AR1.6.** We thank the referee for this comment.

**RC1.7. [In response to previous review item 1.5]**
I appreciate your work, but cannot find this discussion in the revised paper. Where did you include (part of) this discussion?

Let me add a comment for you:
what I was trying to exaplain is that because of this effect (of Fig ii of your reply), your model may be described by "count", the y-axis of Fig ii, instead of the proba p. Let me call "count" C here. Using C as parameter is completely equivalent to using p.
Using C as control parameter, it becomes obvious that as soon as C > 10^5, i.e. 1000 times its baseline value, what you are actually doing is a quasi- extremal dynamics, since you are almost always picking this site.
For lower values of C (in the range p~0.007 I guess C is much closer to its baseline value), you are not doing extremal dyanmics, but since you load all sites at random almost equally, your loading protocol is in effect quite similar to uniform loading.

I just noticed this fact while reading your paper and I think one needs to study this matter carefully in order to compare with other loading protocols.

It is not necessary to have this discussion in full in your paper, a short comment to let the reader realize this fact will be enough.

**AR1.7.** We have incorporated a clearer discussion of this fact in the revised paper. Although we did not include the Figure from our previous response, we discussed how the the effect of $p$ can also be studied further to provide a comparison between the model and existing protocols.

**RC1.8. [In response to previous review items 1.7-1.10]** OK

[for 1.9]: Thanks, this is now very clear when reading the paper, and furthermore one understands why it is important to threshold (relative to interevent time statistics). I learned something new, thank you !

**AR1.8.** We thank the referee for this comment.

**Sandpile-based model for capturing magnitude distributions and spatiotemporal clustering and separation in regional earthquakes**

Rene C. Batac, Antonino A. Paguirigan Jr., Anjali B. Tarun, and Anthony G. Longjas

**Response to Referee Reports**

RC – Referee's Comments

AR – Authors' Response

Referee #3: Stefan Hergarten, stefan.hergarten@geologie.uni-freiburg.de

Sorry for not taking part in the first round of review, although I was asked. Therefore my review mainly refers to the question whether the authors addressed the numerous concerns raised by the two first reviewers appropriately. I feel that they did in general, but in my opinion there are still two important points that require more clarification.

We thank the referee for participating in the review of our paper, and, in the process, reading and commenting on the previous versions of the manuscript and the responses. In our responses below and in the revised manuscript, we attempt to address his concerns.

**RC2.1.** Points 1.4, 2.2 and 2.6 of the original reviews refer to the role of the parameters used for calibrating the model and the relationship to established models in this field. If I understood the model correctly, $\nu = 0.25$ (at $p = 0$) should reproduce the original BTW sandpile model, while $\nu \to 0$ (at $p = 0$) should correspond to the OFC model in the conservative limit. Both cases are characterized by scaling exponents in the event size distribution lower than the values found in this manuscript, and those found here fit much better to real earthquakes. The finding of larger exponents between the two limiting cases is interesting and unexpected, but it should be shown clearly, perhaps with a figure showing the dependence on $\nu$. In this context it should also be made clear that the finding is not a spurious effect of an exponentially decreasing event size distribution.

**AR2.1.** We thank the referee for raising this important concern again.

To clarify the parameters of the model, we note that the driving rate $\nu$ is the value of the external trigger that is added to a specified cell in the grid every time step. In the Bak, Tang, and Wiesenfeld (BTW) sandpile, each cell can have only discrete states $\sigma \in \{0, 1, 2, 3, 4\}$, where $\sigma_{max} = 4$ is an unstable one, i.e. at $\sigma_{max}$, the site will "topple" to redistribute stress to its nearest neighbors. Because of the discrete nature of the BTW sandpile, its $\nu$ can only be discrete, in this case $\nu = 1$, or ¼ of the $\sigma_{max}$. The 2D BTW sandpile produces avalanche size distributions with power-law exponents of around -1.0; the 3D BTW, on the other hand, recovers a power-law exponent of -1.3.

With continuous-state sandpiles, however, the scaling exponent tends to be different from the BTW case, i.e. it is not straightforward to expect the same exponent using the same ratio $\nu/\sigma_{max}$. Lübeck (Phys. Rev. E, 56,1590-1594, 1997), using the Zhang sandpile that corresponds to our $p = 0$ case, presented a trend for the scaling exponents of the avalanche

size distributions for different values of the driving rate, and found that while very small driving rates (« 1/32) tend to preserve the scaling exponent, a hallmark of SOC, the exponents (~1.3) are different from those obtained in the BTW (~1.0). Moreover, for driving rates above 1/32, there is a nontrivial trend for the scaling exponent; the exponents can go higher than the baseline value.

A similar model by Piegari, et al. (Geophys. Res. Lett. 33(1), L01403, 2006) showed the effect of the finite driving rates on the shape of the avalanche size distribution. Their model is another continuous-valued cellular automata with similar rules as ours, although they incorporated the degree of conservation $C$ and assymmetrical neighbor redistribution fractions $f$ (as in the OFC). Their model has shown a crossover from pure power-law to normal distributions as the driving rate is increased. The intermediate power-law regime tends to get steeper as the driving rate is increased; interestingly, their power-law behaviors for very small driving rates also start from exponents higher than the BTW case of 1.0. According to them, "Such a behavior is to be expected since for strong driving rates all internal correlations are washed out." In one of their presented results (Figure 2 inset of their paper), the exponent they obtained for the case of $v = 5 \times 10^{-3}$ (comparable to our $v = 0.001$) approaches 1.6 as the level of conservation is increased. This behavior, in fact, has been verified by our previous work (Juanico, Longjas, Batac, Monterola, Geophys. Res. Lett. 35, L19403, 2008) both in model results (ours was for the conservative case $C = 1$) and in actual sand avalanches in the lab.

As noted in the last part of **RC1.7**, more detailed studies are needed to obtain the correspondence between the proposed model and other existing discrete models of seismicity. Because this may require further analyses, we added a note on how to further establish where the model stands in light of the previous implementations.

ACTION2.1. A shorter version of the discussion above is added into the revised manuscript.

**RC2.2.** The relationship to earthquake rupture area and seismic moment of the model properties (points 1.2 and 1.3) is still not very clear. $A$ should correspond to the rupture area, and $V$ to the seismic moment.

**AR2.2.** [see also **RC1.4** and **AR1.4**] We have revised our view of our results, accounting for these recommendations by the referee. While we still provided the statistical distributions of $A$, in keeping with the previous sandpile based models, we now mention in the text that $V$ is a better parameter for providing and analogy with the earthquake energy $E$.

Provided that the authors can clarify these points and demonstrate that their results on the scaling exponents are nor a spurious effect, I think that the manuscript brings some progress into the understanding of such simples models in the context of earthquakes, so that I would recommend publication then.

We again thank the referee for appreciating the value of our work.

**Sandpile-based model for capturing magnitude distributions and spatiotemporal clustering and separation in regional earthquakes**

Rene C. Batac, Antonino A. Paguirigan Jr., Anjali B. Tarun, and Anthony G. Longjas

**List of Changes**

0. General Changes
   - Exponents reported in decimal values instead of exact fractions [see **RC1.0**]
   - Notation: Magnitudes are now represented by $m$ (instead of $M$) and threshold magnitudes by $m_{th}$ (instead of $m$) in text and figures [see **RC1.5**].
   - Changed the citation of Batac, Acta Geophysica from 2015 (online publishing) to 2016 (print publishing).

1. Section 1: Introduction
   - Fixed some citation issues (\citet and \citep issues)
   - Removed "avalanche energies deemed to be analogous to earthquake energies" [see **RC1.4**, **RC2.2**].

2. Section 2: Model Specifications
   - Added the description of $V$ as one of the quantities being tracked for representing the event size.

3. Section 3: Model Results
   - In the first paragraph, a discussion of the scaling exponents of other continuous-state sandpiles are presented, along with the difference with the BTW exponent [see **RC2.2**].

4. Section 4.1: Energy Distributions and the Gutenberg-Richter Law
   - Explained that the distribution of $A$ is presented in keeping with the earlier models that tracked only $A$ and not $V$ [see **RC1.4**, **RC2.2**].
   - Introduced the parameter $V$ as the better analogous quantity to the earthquake energy $E$ [see **RC1.4**, **RC2.2**]

5. Section 4.3: Temporal Separation of Earthquake Events
   - Separated the discussion of data and model results, for clarity.

6. Section 4.4: Model Advantages and Insights on Empirical Modeling
   - Added a discussion of the implications of the model and comparisons with other models [see **RC1.7**, **RC2.2**].

7. Figures 1-3
   - Made consistent in colors and notations

8. Figure 4
   - Panel (a): Added a power-law with $\beta = 1.45$ as guide to the eye [see **RC1.3**, **RC1.4**]
   - Panel (b): Changed the exponent from $\alpha$ to $\delta$ to avoid confusion ($\alpha$ is already used in the avalanche size distribution scaling) [see **RC1.3**, **RC1.4**].

9. Figure 5
   - Legend entries updated to clearly indicate the data and the shuffled sequences.

10. Acknowledgments
    - Added our acknowledgment of the editor and referees for the useful comments.